# Contemporary Techniques for Remediating Endocrine-Disrupting Compounds in Various Water Sources: Advances in Treatment Methods and Their Limitations

**DOI:** 10.3390/polym13193229

**Published:** 2021-09-23

**Authors:** Kamil Kayode Katibi, Khairul Faezah Yunos, Hasfalina Che Man, Ahmad Zaharin Aris, Mohd Zuhair Mohd Nor, Rabaah Syahidah Azis, Abba Mohammed Umar

**Affiliations:** 1Department of Food and Process Engineering, Faculty of Engineering, University Putra Malaysia, Serdang 43400, Selangor, Malaysia; kamil.katibi@kwasu.edu.ng (K.K.K.); zuhair@upm.edu.my (M.Z.M.N.); 2Department of Food, Agricultural and Biological Engineering, Faculty of Engineering and Technology, Kwara State University, Malete 23431, Nigeria; 3Department of Biological and Agricultural Engineering, Faculty of Engineering, University Putra Malaysia, Serdang 43400, Selangor, Malaysia; hasfalina@upm.edu.my; 4Department of Environment, Faculty of Forestry and Environment, Universiti Putra Malaysia, Serdang 43400, Selangor, Malaysia; zaharin@upm.edu.my; 5Material Processing and Technology Laboratory (MPTL), Institute of Advance Technology (ITMA), University Putra Malaysia, Serdang 43400, Selangor, Malaysia; 6Department of Physics, Faculty of Science, University Putra Malaysia, Serdang 43400, Selangor, Malaysia; rabaah@upm.edu.my; 7Materials Synthesis and Characterization Laboratory (MSCL), Institute of Advanced Technology (ITMA), University Putra Malaysia, Serdang 43400, Selangor, Malaysia; 8Department of Agricultural and Bioenvironmental Engineering, Federal Polytechnic Mubi, Mubi 650221, Nigeria; wazabam@gmail.com

**Keywords:** endocrine-disrupting compounds, occurrences, treatment processes, catalytic degradation, ozonation, adsorption-membrane hybrid process

## Abstract

Over the years, the persistent occurrence of superfluous endocrine-disrupting compounds (EDCs) (sub µg L^−1^) in water has led to serious health disorders in human and aquatic lives, as well as undermined the water quality. At present, there are no generally accepted regulatory discharge limits for the EDCs to avert their possible negative impacts. Moreover, the conventional treatment processes have reportedly failed to remove the persistent EDC pollutants, and this has led researchers to develop alternative treatment methods. Comprehensive information on the recent advances in the existing novel treatment processes and their peculiar limitations is still lacking. In this regard, the various treatment methods for the removal of EDCs are critically studied and reported in this paper. Initially, the occurrences of the EDCs and their attributed effects on humans, aquatic life, and wildlife are systematically reviewed, as well as the applied treatments. The most noticeable advances in the treatment methods include adsorption, catalytic degradation, ozonation, membrane separation, and advanced oxidation processes (AOP), as well as hybrid processes. The recent advances in the treatment technologies available for the elimination of EDCs from various water resources alongside with their associated drawbacks are discussed critically. Besides, the application of hybrid adsorption–membrane treatment using several novel nano-precursors is carefully reviewed. The operating factors influencing the EDCs’ remediations via adsorption is also briefly examined. Interestingly, research findings have indicated that some of the contemporary techniques could achieve more than 99% EDCs removal.

## 1. Introduction

The detection of endocrine-disrupting compounds (EDCs) as contaminants in the environment has drawn the significant interest of researchers during the past few years, owing to their potential human and environmental threats [1]. Several chemicals (some illicit and some still in circulation) have been considered as EDCs [2]. The increasing accumulation of more EDC micro-contaminants in natural waters is mainly attributable to the advancement and rapid expansion of chemical technologies for organic production and processing [3].

These contaminants can infiltrate directly into the aquatic environment via effluent outflow and indirectly as runoff, yet the main carrier of EDC contaminants to the freshwater bodies is via treated and raw urban effluent release into water bodies [4,5]. Moreover, even most of the treated potable water resources may be polluted through deep-well injection of the effluent and surface water outflow [6]. This shows that even treated water is not absolutely free from the EDC contaminants [7,8]. The persistence of EDCs in water even at a trace concentration is notably dangerous to the health because of its ability to cause metabolic and reproductive disorders; therefore, the need for efficient management of EDCs contained in effluent before discharge is indispensable [9,10].

The management of effluent discharges emanating from various sources, such as pharmaceutical compounds, pesticides, personal care products, and similar compounds, has received significant attention in many countries [11,12]. This ensures considerable control, even though more stringent regulations are still required for better management [13]. Various studies have indicated that EDCs are ubiquitous and can frequently be found in almost all water sources, namely surface waters, groundwater, municipal water, treated and untreated wastewater treatment plant (WWTP) effluent, and finished drinking water, globally [14,15]. Primarily, the most practiced management technique is a conventional treatment. In this vein, several reports have indicated that the conventional treatment approach is inefficient in the elimination of EDC contaminants from water [8,16,17,18]. This is because several EDCs are non-biodegradable in nature or have poor biodegradability and strong chemical cohesion in the environment [19]. For instance, about 41 and 40 EDC pollutants were found in the treated effluent and environmental waters at the downstream and upstream of wastewater treatment facilities (WWTF), respectively [20]. Notably, among these, higher proportions of BPA (239.0 ng/L; 396.4 ng/L), diclofenac (467.7ng/L; 1461.5 ng/L), carbamazepine (157.1 ng/L; 279.5 ng/L), and ibuprofen (153.3 ng/L; 312.1 ng/L) were recorded in the effluent of both upstream and downstream of WWTF. Analogously, Mailler et al. [21] reported that NP and BPA were detected in the treated effluents of wastewater treatment plants, ranging between 100 and 1000 ng/L, as well as higher proportions of artificial sweeteners, close to 1000–10,000 ng/L. Besides, effluent discharge from municipal wastewater was classified as the major source of EDCs in the rivers in China [22]. Moreover, Lin [7], in his study, established that some selected EDC contaminants such as DEET and TCEP are relatively resistant to the conventional treatment process. This trend was corroborated by Carmona et al. [23], who established that PPCPs compounds ranging between 6.72 to 940 ng/L were discovered in an effluent discharge after the conventional wastewater treatment process.

Despite the tremendous efforts of researchers to improve the existing conventional treatment technologies as highlighted above, complete removal of the EDCs remains a challenge. Thus, several studies in recent years, reviewed by several authors, have investigated and focused mostly on the occurrence, fate, transport, and elimination of endocrine-disrupting compounds, as well as treatment techniques for the removal of EDC micropollutant from various water sources [12,13,24,25,26,27], demonstrating major concern about them.

No attempts have been made to provide an exhaustive review of the adverse impacts of EDC contaminants on humans and ecosystems, as well as the removal of these bio-persistent contaminants using various recent advances in the treatment techniques and their inherent limitations, despite the significance and relevance of this subject.

To develop a more efficient treatment process, a good understanding of the EDCs and the existing treatment approaches, recognizing their merits and demerits, is critical.

The aim of this paper is to provide a critical review of these EDC contaminants and the existing treatment methods and their inherent drawbacks. The principal working factors controlling EDCs removal via adsorption technique was also carefully discussed.

### 1.1. Nature and Classification of EDCs as Pollutants

According to the environmental route to EDCs, almost 686 compounds are classified as EDCs, which are further divided into seven broad groups, namely consumer products, farming and agricultural, industry, intermediates, natural sources, medicine, and health care contaminants (Figure 1). In addition, the seven broad groups of EDCs were further categorized into 48 sub-groups (Figure 1). Notably, this environmental-source-based categorization of EDCs is overlapping; specifically, a given EDC may fit into multiple sub or broad groups [28]. Bisphenol A (BPA), phenols, and phthalates are categorized as EDCs since they can hinder hormone or endocrine systems [29]. Though several compounds are classified as endocrine-disrupting compounds, steroid estrogens, particularly 17β estradiol (E2), 17α- ethinyl estradiol (EE2), as well as estriol (E3), have been extensively researched due to their higher estrogenicity at trace concentrations (µg/L& ng/L) and their detection in different environmental matrices, particularly surface, ground, drinking water, and effluent from sewage treatment facilities [30,31]. In addition, EDCs can be natural androgens and estrogens, synthetic androgens and estrogens, phytoestrogens, engineered nanomaterials, pesticides, pharmaceuticals, personal care products, drugs of abuse, as well as other industrial chemicals with high propensities to stimulate harmful impacts on the endocrine systems of humans, fauna, and available water resources [12,32].

### 1.2. Sources and Occurrences of EDCs as Pollutants

EDC pollutants find their way into the water systems via different routes, including human excretion (sewage), landfill leachate, industrial discharge or drain water, and wrongful disposal [20]. Besides, sludge and sewage of livestock, and municipal, hospital, and industrial wastewater treatment facilities are the major entry routes of microcontaminants into the environs [33].

Basically, the pathways for EDCs entering the environment, specifically into the receiving waters, can be categorized as point source (namely industrial wastewater, municipal sewage, landfill) and non-point source (underground contamination, wash off), as illustrated in Figure 2. Point source contributes to the majority of EDC pollution, particularly discharges via wastewater treatment [34], since the major reference point of these pollutants comes from untreated and treated sewage effluents, as well as direct discharge to rivers [10]. According to Cao et al. [35], most common sources of EDCs emanate from the compounds produced by plants and animals, pesticides, and detergents, and leach out of plastics. Presumably, contamination of surface and groundwater is also due to the significant dependence of food production on pesticide chemicals, via various point (traceable) or non-point (disperse) pollution sources [36].

Yet, there are diverse sources of environmental pollutants, such as EDCs in water bodies, which have generated acute problems, since water resources serve as a sink for various contaminants [37]. This implies that the aquatic environment (such as seas, streams, rivers, and groundwater) becomes vulnerable to various detrimental impacts of most micropollutants.

### 1.3. Adverse Effects of EDCs

A considerable amount of scientific literature has reported various deleterious impacts of EDCs on the environment, and their propensity to distort stability in the ecosystem. Recently, there has been a rise in the number of investigations that have highlighted and described various health consequences linked with endocrine-disrupting compounds viz interference with the endocrine system of humans and animals by impacting the synthesis, release, transport, metabolism, and excretion of hormones in the body. They generate their consequence by antagonizing, mimicking, blocking, and altering the normal function of the hormone system and the endogenous steroid stages through modifying their metabolism or synthesis rates in humans, thereby resulting in severe impacts, in particular irregular reproductive development, cardiovascular changes, metabolic disease (particularly obesity, diabetes, etc.), reduction of sperm reproduction in humans which results in low fertility, thyroid and adrenal gland dysfunctions, immune and neurological diseases, developmental dysfunctions throughout the fetal period, stimulation of breast cancer in women, development of testicular and prostate cancer, decline in reproductive fitness of men, and increased threat to humans [38,39,40,41]. 

Table 1 presents a concise summary of the reported adverse effects of the EDC contaminants. As indicated in Table 1, EDCs have also be linked to altered behavior and obesity in children, reduced gonadal development and viability, and alteration of the physiological status of humans and wildlife [42,43]. Notably, exposure to EDCs by humans and wildlife results from the intake of chemicals through foods and drinking water consumption, which results in biomagnification and bioaccumulation, particularly in species at the highest level of the food web [44]. Furthermore, research findings have also reported negative impacts of EDCs on animals, as they affect the hormonal systems of organisms, binding to receptors in animals and mimicking the activities of estrogen, obstructing the normal action of the endocrine system and inciting reproductive syndrome, feminization, and carcinogenesis in numerous wildlife animals; they affect the synthesis release, transport, and interface with female estrogen and disrupt the reproductive growth and behavior of animals; and they interfere with the delicate balance of the endocrine system of animals, altering the physical status in wildlife and natural hormone activities and threatening the reproductive biology and health of an animal populace [42,43,45,46,47].

In addition, numerous studies have extensively reported various irregularities observed in the marine environment due to the presence of EDCs, which include: biomagnification and bioaccumulation in the aquatic ecosystem, intersex and skewed sex ratios, reduction in fish fertility, modified gonadal growth (intersex and imposex), anomalous blood hormone levels, initiation of protein expression and vitellogenin gene in juveniles and male fish, masculinization and feminization, disruption of the reproductive mating behavior of fish, intersex in sucker fish downstream of a sewage plant effluent, hermaphroditism, and decreased fertility and fecundity [45,47,48,49,50]. Exposure to EDCs has also been reported to pose a potential risk to the water quality and the ecosystem because EDCs can undermine water quality, increase adverse ecological impacts, and be considered as environmental pollution with relatively elevated biological activity [51,52,53]. These findings justify the need for critical studies of the EDC contaminants in conjunction with the existing treatment methods and their drawbacks, as well as the operating conditions. Primarily, this could serve as useful information for improving the existing treatment processes to provide more efficient remediation performance.

## 2. Treatment Processes in Removing Endocrine Disrupting Compounds

The emergence of unregulated micro-contaminants, such as endocrine-disrupting compounds (EDCs), created the need for effectual treatment technologies to remediate the concentration level [37]. Recently, several approaches for the elimination of EDCs from wastewater, including potable water, have been reported. These include the conventional treatment method, adsorption process, biological treatment based on enzymatic degradation, photocatalysis degradation, ozonation and oxidation processes, use of membrane filtration technique, and hybrid systems [24,43,71,72].

### Conventional Treatment Process

The conventional treatment process comprises three major phases, namely primary (or mechanical), secondary, and advanced phases for the remediation of EDC contaminants from water sources. The initial primary phase is configured to eliminate the suspended, gross, and floating solids from raw wastewater from its source. It also involves screening to confine solid objects and removal of suspended solids through sedimentation by gravity [71]. The secondary treatment contains activated sludge, which employs an aeration tank or dispersed-growth reactor containing microorganisms (consuming the organic matter and converting it into carbon dioxide (CO_2_), water, and energy to enhance reproduction and development), mixed liquor, and a suspension of wastewater. The constituents of the aeration tank are agitated vigorously by the aerator, which then supplies required oxygen to the biological suspension [72]. The trickling filters in the secondary treatment phase serve as a support media where wastewater is applied intermittently or continuously over the media, such that, as the water flows, the microbes become linked to the media and build a fixed film. In this context, the organic matter in the wastewater dissolves into the film, where it is metabolized [73]. The conventional treatment technique has remained the most extensively utilized treatment process for decades and is widely considered to be very efficient in handling reclaimed water by eliminating the mass of microbial contaminants and chemical compounds. Hence, the efficiency of the conventional treatment process has been investigated by researchers to assess its effectiveness in removing EDC contaminants from water. Table 2 presents a summary of findings regarding the existing conventional treatment approaches.

Summary of published information on the elimination of EDCs from different water sources using various treatment methods.

For instance, Ye et al. [75] studied the treatability of EDCs using conventional wastewater treatment facilities. Over 85% removal was recorded during this study. It was observed that bisphenol A (BPA) was detected in abundant proportions in the effluent and sludge, with the highest levels being 1210.7 ng/L and 2470.4 ng g^−1^ dw, respectively. However, the major constraint of this study was the potential risk to the public health and ecosystem due to the residual proportions of target EDC pollutants in the treated effluent, together with longer HRT and SRT. Zhang et al. [76] applied conventional techniques to remove eight EDCs in a sewage treatment facility. The overall removals achieved ranged between 16.9% and 94.4%. The authors found that both primary and biological treatment units could remove target EDCs. However, the major shortcoming of this study was the presence of a higher concentration of target EDCs in the effluent (treated water), with consequent estrogenicity which could result in elevated ecological hazards to the receiving surface water and marine environment. In another study, Samaras et al. [74] examined the occurrence and fate of five EDCs using the conventional treatment technique and anaerobic sludge digestion. The results of their study showed that removal efficiencies of 39–100% were achieved for the pharmaceutical compounds DCF and IBF during conventional treatment, while IBF and NPX recorded over 80% removals during sludge anaerobic digestion. However, two major limitations observed from this study were the average removals of NP1EO (˂55%) and higher concentrations of triclosan and nonylphenol detected in the treated wastewater in most cases, which could constitute a severe ecological menace to the aquatic environment. 

Qiang et al. [63] conducted a comparative investigation on different wastewater treatment approaches (stabilization pond, constructed wetland, activated sludge, and micropower reactor) for the elimination of EDCs from 20 rural wastewater treatment facilities in Zhejiang province, China. A maximum removal of 70% was attained during the study. The authors found that the performance of a centralized activated sludge process surpasses the other three decentralized processes. However, this study had several drawbacks, including poor removals of target EDCs by stabilization pond, unstable performance of decentralized processes, and negative impacts of effluent discharged from the treatment plant on the quality of the receiving river because of micro-pollution; in addition, the efficiency of removal largely depends on the sampling season and specific wastewater treatment technique. 

Based on the above findings, the removal efficiencies of a conventional wastewater treatment process vary depending on the specific treatment method and sampling season, and the characteristics of the microcontaminants. The major removal mechanisms for EDCs removal during secondary treatment are sorption, biodegradation/biotransformation, and chemical reaction [74,77]. Owing to its wider application, the performance of the conventional activated sludge (CAS) process relies on the nature of the microbial community and physicochemical characteristics of EDC pollutants. Notably, the most critical working conditions influencing the performance of CAS include the sludge retention time (SRT), hydraulic retention time (HRT), temperature, and disinfection process [74,78]. Additionally, prolonged HRT could facilitate the elimination of more recalcitrant contaminants, and a longer SRT could allow a greater diversity of microbes [79]. The standard SRT during the CAS technique is between 7 and 20 days, and HRT ranges between 2 and 24 hrs, with the proportion of biomass ranging between 3 and 5 kg m^−3^ [77]. 

Several researchers [8,80,81] have distinctly established that the conventional treatment technique is inefficient in eliminating emerging EDC contaminants. Importantly, some of the identified inherent drawbacks of the CAS process include strong resistance of some EDCs to the CAS degradation process, presence of residual EDCs contaminants and pharmaceutical compounds in the treated effluent which might cause several health problems or present potential hazards to the ecosystem, too many modular units, prolonged hydraulic retention time (HRT) and sludge retention time (SRT), higher footprint, low pathogen removals, and high maintenance requirement. 

Efficient techniques for eliminating EDC pollutants from water and contaminated sites are required. This substantiates the need to employ more advanced treatment technologies.

## 3. Contemporary Techniques for the Removal of EDCs from Various Water Sources

Innovative techniques for wastewater treatment are essential to exterminate pollution and may also perhaps enhance separation procedures or contaminants destruction. These techniques include advanced oxidation methods (photocatalytic and catalytic oxidation), membrane separation, MBR, adsorption, and hybrid systems. These techniques can be successfully applied to eliminate contaminants that are incompletely removed by conventional systems, such as suspended solids, heavy metals, colloidal substances, biodegradable organic compounds, phosphorus and nitrogen compounds, microorganisms, and dissolved compounds, thus enabling reusing of residual water [3].

### 3.1. Catalytic Degradation of EDCs

The catalytic degradation process involves stimulating the rate of degradation of the EDCs [82]. The enhancement of the degradation rate may be achieved via the presence of photo radiation or organic enzymes [83]. Primarily, the catalyst offers a choice of reaction path with lesser excitation energy compared with the non-catalysed mechanism. Typically, the catalyst reacts to build a temporary intermediate in catalysed mechanisms, which subsequently rejuvenate the original catalyst in a virtuous circle [80].

### 3.2. Photo-Catalytic Degradation of EDCs

Generally, the photocatalytic degradation of EDC contaminants requires the stimulation of photoreactions under a combined influence of light (solar irradiation or UV) and a catalyst [19,81]. It involves multiple steps (such as diffusion and adsorption of EDCs, chemical reactions, desorption of intermediates, and removal of the product from the interface) to complete the process. The reaction products (intermediates) of these steps ultimately constitute the end products during the last stage. Usually, the desired end products of a completed photocatalytic degradation process are H_2_O and CO_2_. Detection of these reaction intermediates would offer additional insight into the mechanism involved in the degradation technique and would facilitate the degradation pathway. Thus, the efficacy of a successful and higher photocatalytic process is based on the generation of HO^•^ radicals [19].

The performance and degradation rate of a photocatalytic process hinges on several working conditions that determine the elimination of EDCs in water. These include light intensity, wavelength, presence of organic and inorganic compounds, reaction temperature, catalyst loading, concentration and chemical structure of the contaminants, initial concentration of the substrate, solution pH, and dissolved oxygen [19,84]. Few studies have explored photocatalytic and enzymatic degradation processes to eliminate EDCs from different water sources (Table 3). Furthermore, the photocatalysis process has been considered as a promising technique for degrading EDCs, with no secondary contamination, moderate reaction medium, and better energy-saving [85].

For instance, Frontistis et al. [86] examined the photocatalytic degradation of 17α-ethynyl-estradiol using ZnO under simulated solar radiation. The results of the study indicated that EDCs were efficiently treated with rapid degradation via first-order kinetics. However, some of the shortcomings of this study were the detection of estrogenic compounds in the photocatalyzed effluent, retardation of the degradation process by the copresence of organic and inorganic matter in the secondary effluent, and overall remove rate of effluent estrogenicity being very low. A similar study from Zacharakis et al. [43] investigated the degradation of bisphenol A (BPA) under synthetic solar irradiation in the presence of TiO_2_ or ZnO catalysts immobilized onto glass panels. It was discovered that the use of TiO_2_ or ZnO in the photocatalytic degradation with low energy demand is a promising and efficient technique to eliminate EDCs from water. However, low degradation in treating wastewater, slight hindrance of BPA degradation due to the presence of EE2 (particularly at 50 and 100 µg/L), and retardation of the degradation process owing to the copresence of inorganic and organic matter have limited the application of this treatment process.

### 3.3. Enzymatic Degradation

Phytoremediation (enzymatic degradation) is another novel remediation and a promising technique for the elimination of EDCs and other similar chemical compounds in wastewater. Researchers have identified several micro-organisms as critical factors to proceed with the EDC phytoremediation process, and the most widely applied ones are fungal, bacterial, and algal strains, as well as mixed cultures [92]. Enzymatic degradation also depends on the microorganism activities, although the degree of degradation has a strong correlation with several environmental factors, such as pH, nutrient, and temperature [93]. Some of these bio-enzymes include oxidoreductases: laccases, tyrosinases, polyphenol oxidases, manganese peroxidase, lignin peroxidase, horseradish peroxidase, and bitter gourd peroxidase. Studies have collectively indicated that apart from the environmental factors, quite a few redox mediators, additives, and surfactants could better enhance the enzymatic oxidation process [94]. Table 3 presents the recent findings on the use of enzymes and their treatment conditions for removing EDCs. 

Macellaro et al. [88] examined the degradation of five different EDCs using four distinct fungal laccases, subject to the availability of both synthetic and natural mediators. The results obtained from this study revealed that all laccases could oxidize different EDCs, with bisphenol A (BPA) exclusively oxidized under all conditions tested. In addition, mediators remarkably increase the performance of enzymatic treatment and enhance the degradation of substrates refractory to laccases oxidation. Two main possible limitations of this study were the tedious nature of the experiment procedure and challenges in adapting enzymes capable of eliminating the target compounds with an affinity constant of the same order of magnitude concerning the typical proportions of EDCs in the surroundings.

Studies have also reported the degradation of hormones, phenolic compounds, and some other EDC contaminants using oxidative enzymes in aquatic plants (floatable and submerged plants) and fungal laccases from synthetic and natural wastewaters [87,90] during batch and continuous processes. Their findings showed that the degradation of target phenolic EDCs via oxidative enzymes in the aquatic plant was successful, and removal of hormones and EDCs using laccases is feasible at a trace concentration (2.8 ABTS U/L). However, the formation of by-products during enzymatic degradation, the inability of free laccase to effectively remove various EDCs in more complex matrices, more extended treatment period (100 days), and tedious complex experimental procedures are the major constraints of these studies. Similarly, Haugland et al. [89] investigated the laccase removal of 2-chlorophenol and sulfamethoxazole in urban sewage. The experimental results showed that excellent removal was achieved without acetosyringone by the natural enzyme mix, and sulfamethoxazole was adequately removed from secondary effluent without mediators. However, this study failed to define the removal mechanisms and conditions to maximize the removal rate with consequential potential by-products production. Besides, this process is not economically viable due to a lack of provision for the onsite production of laccase, recycling, and immobilization of the enzymes for multiple uses.

Scientific studies have proven that phytoremediation (enzyme degradation) is a promising emerging technology and better than the conventional technique in terms of environmental impediments, price-to-performance ratio, and affordable treatment for the remediation of EDCs from wastewater, due to the utilization of plants in degrading chemical contaminants [95]. It was observed that the removal capacity of enzymatic degradation strongly depends on the unique enzyme systems, fungal species, and properties of EDCs. 

Some major constraints of the enzymatic degradation process are the following: significant purity is indispensable for enzyme crystallization, production of cross-linked enzyme aggregates (CLEAs) is involved in laccase precipitation, and there are challenges associated with combining the treatment and immobilization into a single process [96,97]. Additionally, the immobilization process usually presents poor laccase regeneration [98]. The addition of mediator hydroxy benzotriazole to the crude enzyme extract led to an improvement of some phenolic as well as non-phenolic EDCs degradation. However, this was accompanied by higher residual toxicity in the treated media [99]. The degradation of EDCs by enzymes often involves some challenges because of their product inactivation and the recalcitrant nature of the EDCs. Therefore, developing full-scale treatment operations using oxidoreductive enzymes is not viable due to their potential deactivation and slow process kinetics [93]. Hence, there is a need for future studies on laccases recovery, the viability of oxidoreductive enzymes, and potential deactivation to improve the immobilization process.

### 3.4. Removal of EDCs by Membranes

Membrane technology is the most extensively applied physicochemical separation technology for the removal of salt and microbes from water [100,101,102]. Membrane processes have been productively utilized in difficulties relating to unavailability of fresh and clean water and could remove EDCs and natural organic matter (NOM) from both wastewater reuse and drinking water [103,104]. This could be achieved due to its unique characteristics, including energy efficiency, compactness, high throughput, and cost-effectiveness [105].

Essentially, pressure-operated membrane processes are described and classified into four major classes, mostly based on the pore size and operating pressure exerted: microfiltration, ultrafiltration, nanofiltration, and reverse osmosis [106,107].

Moreover, polyvinylidene fluoride (PVDF), polyacrylonitrile (PAN), polyethersulfone (PES), cellulose acetate (CA), and polysulfone (PSF) are the most frequently applied polymer materials in membrane purification for water treatment [108]. Among these, PVDF is the most favored and broadly employed polymeric membrane and has drawn growing interest in recent years from manufacturers and researchers. This is because PVDF polymer has exhibited unique and promising characteristics that make it an effectual and superior candidate to reject EDC contaminants from water. These include exceptional aging resistance, outstanding mechanical strength and thermal stability, and good chemical resistance, which are central for the practical application of membrane technology [109]. In addition, PVDF shows acceptable processability for fabricating flat sheet, hollow fibre (HF), and tubular membranes, and it is dissolvable in numerous conventional solvents, such as N, N-dimethyl acetamide (DMAC), dimethylformamide (DMF), and N-methyl-2-pyrrolidone (NMP) [110]. The chemical and physical characteristics of the material could strongly affect membrane performance [100], since the ideal membrane is one that can yield a high flux with zero fouling or clogging and that is chemically stable and resistant, physically durable, nonbiodegradable, and low cost. Table 4 presents a summary of some research findings on the application of membrane treatment technique in eliminating EDCs pollutants.

Several studies have reported that ultrafiltration (UF) membrane could moderately eliminate EDC pollutants from different water sources [111,112,113,114,115]. Principally, UF and MF membranes have been regarded as a promising solution to minimize operating costs since they can be operated at reduced pressure (1–3 bar) and produce more permeate in a shorter period [116]. Nevertheless, it should be noted that microporous membranes cannot eliminate EDCs as efficiently as dense membranes by size exclusion. Thus, consideration should be taken of other mechanisms, such as adsorption and electrostatic interaction. The efficiency of UF and MF membrane in eliminating EDCs could reach that of NF and RO membranes, provided that the adsorption mechanism plays a crucial role in retaining EDCs by adsorbing onto the internal pore walls and membrane surface, since adsorption is the principal mechanism for the retention of micropollutants via UF membranes [117,118]. Similarly, Hu et al. [119] investigated the fouling disposition of simulate effluent and the influence of membrane fouling and working pressures on the rejection of selected EDCs during UF membrane filtration tests. The results of the study showed that the fouled membrane could eliminate 10–76% of some target compounds. These results showed that the fouled membrane could enhance EDCs removal from water. The authors suggested that 50kPa may be efficient to achieve better EDCs removal with suitable flux. 

A study conducted by Bing-zhi et al. [120] examined the rejection of bisphenol A (BPA) using polysulphone (PS) membrane. The results revealed that the adsorption capacity of BPA towards the membrane is dependent on the material, which has an excellent removal rate when using polysulphone. The authors concluded that BPA retention is severely affected by solution pH, with a sharp decline in retention when the pH exceeds its pKa value. It was concluded that sorption onto the membrane could be energized by both physical and chemical interactions in terms of hydrophobic adsorption and hydrogen bonding, respectively. There may be some possible limitations in this study, as polysulphone membrane is highly susceptible to adsorptive fouling, a large number of contaminants are accumulated, and changes in the feed solution matrix led to leakages. Accordingly, the combinations of different membrane processes, such as MF, UF, RO, and NF, is crucial to strengthening the elimination of different EDCs [26].

Yüksel et al. [121] evaluated the rejection of bisphenol A (BPA) from model solutions using selected RO and NF membranes. Excellent rejection (≤ 98%) of BPA was achieved with three polyamide RO membranes. Despite these significant removals, high energy demand and too many modular units (membranes) remain the major drawbacks of this study. Hence, the application of this process is not economically feasible, especially in a full-scale application. Zielińska et al. [53] combined MF and NF to remove EDCs from biologically treated wastewater. In this study, it was discovered that the two processes achieved complete removal of BPA at an initial proportion of 0.3 ± 0.14 mg/L, and a removal efficiency of 61–75% was recorded for the NF membrane. The authors concluded that the MF membrane appears as a favorable panacea for the subsequent treatment of wastewater containing BPA and could be applied at lower transmembrane pressure (TMP) than NF. The two major limitations observed in this study were a decline in the filtration capacity due to fouling and quick fouling in the MF membrane, thus reducing the removal efficiency from 37% to 24%. Interestingly, a higher removal efficiency of 97% was reported by Al-Rifai et al. [122] when MF and RO were combined in treating EDCs. Yet, BPA at a concentration of 500 ng/L was discovered in the effluent after the treatment process and a higher energy demand was required.

Often, adsorption may override electrostatic repulsion as a removal mechanism. This was evident in a study undertaken by Yoon et al. [104] to investigate the rejection of EDCs of different physicochemical properties using NF and UF membranes in a filtration process. Results revealed that 30 to 90% of EDCs could be eliminated through the NF membrane, compared to only less than 30% observed in the UF membrane. Both steric hindrance and hydrophobic adsorption were the removal mechanisms for the removal of EDCs in the NF membrane, while the UF membrane relies entirely on the hydrophobic adsorption mechanism for the removal of hydrophobic EDCs. The authors concluded that the transport phenomenon associated with adsorption is driven by the membrane material and water chemistry conditions.

Recently, nano-composite membranes have been taken into more consideration due to improved characteristics and more removal efficiency and permeate flux. Recently composite membranes have been manufactured from several constituents to combine the component materials’ strength in the final product. Usually, one material represents the active surface, while the other represents the support layer [12]. Nano-composite membranes are manufactured by adding nanoparticles in the membrane formulations. Several studies have proven that all the membrane processes could reject EDCs from water. However, better rejection of EDCs could be achieved via the application of high-pressure-driven membranes, particularly RO, NF, and FO, considering size exclusion mechanism. However, high energy demand and associated costs in RO and NF have reduced the wider application of these systems [100]. Particularly, a membrane with larger pores, particularly UF and MF, can also be considered if the principal removal mechanisms associated with the process are electrostatic repulsion and adsorption.

Hence, the application of UF and MF membranes with relatively lower energy demand, due to their lower pressure and closely connected with their low cost, deserves more attention for the treatment and rejection of EDCs, since the UF system has a practicable market demand in advanced water treatment. Its efficacy could be robustly enhanced via alteration of the membrane surface to significantly remove EDCs from water and mitigate fouling, without undermining membrane permeability and flux. Mostly, nanoparticles (NPs), including titanium dioxide (TiO_2_), silica (SiO_2_), and iron oxide (Fe_3_O_4_), have been exploited for expanding the properties of UF membranes [123]. Substantial removal (˃98%) of BPA and other contaminants was achieved when nanoparticles were integrated in the membrane matrix in numerous studies [124,125,126,127]. For instance, Zahari et al. [128] acknowledged that the integration of magnesium oxide (MnO_2_) NPs and polyvinylpyrrolidone (PVP) in PVDF membranes could substantially improve the membrane properties and achieve a complete (100%) removal of BPA. It was noticed that the membrane also exhibited excellent reusability for BPA rejection. Anan et al. [115] investigated the capacity of thin film composite UF membrane immobilized with TiO_2_ NPs on the rejection of BPA. It was observed that TiO_2_/PA membrane could remove nearly 99% of BPA from the synthetic solution. It was also reported that the hydrophilicity of the membrane was enhanced due to the presence of TFC. Similarly, Wang et al. [129] in their study investigated the removal of BPA using Fe-doped PSF/TiO_2_ composite UF membranes. BPA removal efficiency of 90.78% was achieved in 180 min with improved self-cleaning ability and mechanical capacity. Nasseri et al. [130] optimized the operating conditions for a nano-composite PSF membrane incorporated with graphene oxide. It was noticed that the maximum BPA removal of 93% was achieved at a pressure of 1.02 bar, 10.6 min, and pH 5.5.

To recap, membrane filtration technique could remove EDC pollutants from water, as highlighted earlier. However, the major drawback of this approach is membrane fouling. Interestingly, this anomaly could be addressed through pre-treatment membrane modification via nanoparticles, air sparging, and optimization of process parameters, among other things. Recent studies have indicated that membranes incorporated with nanoparticles could achieve substantial retention of EDC pollutants. Hence, membrane filtration can synergistically be integrated or combined with other water and wastewater treatment technologies to improve water quality.

### 3.5. Removal of EDCs by Ozonation and Advanced Oxidation Processes (AOPs)

Advanced oxidation processes (AOPs) are generally applied for the elimination of persistent and recalcitrant EDC constituents from municipal and industrial wastewater. In this context, AOP techniques can become very favourable methods for purifying wastewater comprising hardly biodegradable or non-biodegradable organic compounds with excessive poisonousness [3]. The AOPs can be successfully applied in wastewater purification to destroy the persistent EDC contaminants, the oxidation procedure being controlled by the very strong oxidative potential of the HO^•^ radicals produced into the reaction medium by various mechanisms [134]. AOPs are extensively identified as techniques that employ strong radical oxidants (such as ^·^OH, SO_4_^–^) to fast-track or accelerate the removal of several organic pollutants from different water matrices [135]. Notably, ^•^OH is one of the most exceedingly non-selective and reactive radical species existing in AOPs, with a standard reduction potential of 2.8 V vs. standard hydrogen electrode (SHE) [136]. These processes involve the generation of strongly reactive oxidizing hydroxyl radicals (HO^−^) species, such that the generation of ^•^OH could be enhanced in the presence of H_2_O_2_, ultraviolet, and Fenton reagent [137]. AOPs can be employed to oxidize contaminants partly or completely, typically via several oxidants. Photocatalytic and photo-chemical advanced oxidation processes including UV/TiO_2_, UV/H_2_O_2_, UV/H_2_O_2_/O_3_, UV/H_2_O_2_/Fe^2+^(Fe^3+^), UV/O_3_, and UV/H_2_O_2_/TiO_2_ can be utilized for oxidative degradation of EDC contaminants. A complete mineralization of the EDC contaminants is not essential, as it is more valuable to convert them into biodegradable aliphatic carboxylic acids succeeded by a biological process [138]. The preferential utilization of H_2_O_2_ (oxidative agent) and HO radicals producer is evidenced by the fact that the hydrogen peroxide is simple to store, transport, and utilize, with an efficient and safe procedure [134].

Ozonation and AOPs are powerful redox techniques which exhibit remarkable advantages over the conventional treatment process, particularly small footprint, higher degradation rates, and non-selective removal of non-biodegradable persistent refractory compounds that could not be treated by the conventional treatment process [10]. 

Besides, these processes allow decontaminating effects which are crucial for water reusability applications due to direct human contact, such as household reclamation applications [139]. Ozone can degrade organic pollutants directly and indirectly through the generation of a reactive oxidizing agent (^•^OH). The aim of AOPs as a pre-treatment process, either singularly or with supplementary processes, is to enhance the qualities of the conventionally treated effluent and to achieve deactivation of pathogens not treatable by conventional approach [10]. However, several EDC pollutants are susceptible to both ozone and AOPs (particularly carbamazepine and naproxen), while some are simply dependent on ^•^OH (namely meprobamate and atrazine) [140].

Notably, the most frequently applied AOPs to eliminate EDCs from various water matrices comprise of ozonation (catalytic), heterogenous photocatalysis using UV light source, Fenton and photo-Fenton processes, electrochemical oxidation, or a combination of any of the processes [141].

Different catalysts have been identified for catalytic processes subject to the reaction procedure, involving metal oxides (Zn, Mn, Ti, Bi, Cu, and Co, etc.), noble metals (such as Pd, Ir, Pt, Rh, and Ru), or metal-free carbonaceous material (viz., activated carbons, graphite, carbon fibres and foams, carbon nanotubes, and carbon xerogels) [12]. However, the most widely utilized catalyst in ozonation and other advanced oxidation processes is titanium dioxide (TiO_2_) [10]. Amongst the various photocatalysts, TiO_2_ has been demonstrated to be a promising and favourable semiconductor photocatalyst in advanced oxidation processes and heterogeneous photocatalysis due to its low cost, availability, stability, non-toxicity, unique photocatalytic efficacy, and its potential applications in water and wastewater management [142]. Comparatively, degradation of EDCs via a solar photocatalytic approach has not been sufficiently explored, despite it being a promising technique with unique characteristics such as zero secondary contamination, benign reaction condition, facile procedure, and low energy demand [143,144,145,146,147]. Table 5 presents a summary of the recent advances in AOPs applications for remediating EDCs.

Yang et al. [59] examined the removal efficiencies of selected EDCs spiked in a wastewater matrix using ferrate Fe (VI) treatment technology. Study results showed that the Fe (VI) process eclectically oxidized electron-producing organic EDCs, particularly phenols of estrogenic compounds. However, there are two major limitations to this study: firstly, almost thirty-one (31) EDC contaminants were discovered in the effluent of wastewater plants, with proportions ranging between 0.2 ± 0.1 to 1156 ± 182 ng/L; and secondly, this treatment method could not eliminate triclocarban and three androgens.

Zhang et al. [153] studied the application of the Mn-Fenton process enhanced with microwave for the elimination of bisphenol A (BPA) in the water. Significant removal efficiency of BPA (99.7%) was achieved with 34.0 mg/L of H_2_O_2_ concentration, 2.7 mg/L of Fe^2+^ ion concentration, and 100.0 mg/L initial BPA concentration at pH 4. The authors pointed out that the complete removal of BPA can be achieved at minimal initial proportions. However, the optimal pH of 2.5–4.0 used in this study could lead to the production of intricate multi-nuclear complex metals in the secondary sludge which are not suitable for discharge in this state. Zhang et al. [152] found that the ozonation process with 9 mg/L initial ozone concentration at pH 6 could effectively degrade EDCs. In addition, Muz et al. [148] reported that greater than 99% removal of selected EDCs contaminants from the sludge was achieved after the fourth-day pulse ozonation process when 1.1 mg O_3_/L ozone dose was applied. This result showed that EDCs were destroyed to an undetectable proportion in the sludge after ozonation. The authors also found that a 1.1 mg O_3_/L spike was suitable to achieve almost complete elimination of EDCs in the sludge. However, the high cost of ozone and the formation of toxic by-products were the major drawbacks of this study.

Furthermore, Dudziak et al. [149] revealed that removal efficiency of pharmaceutical EDCs compounds greater than 96% was achieved by combining UV/TiO_2_ in the advanced oxidation process. Recently, Saggioro et al. [150] evaluated the degradation of three EDCs from the sewage matrix by incorporating ultraviolet radiation with chlorine using 2 mg/L chlorine concentration. Almost complete degradation was observed with almost 99% of EDCs significantly degraded within 3 min. Yet, there may be some possible constraints in this study due to the formation of chlorate by-products disinfection, which relies on competitive reaction, decomposition rate, and formation of toxic by-product, which could necessitate the need for additional treatment [13,154].

Research findings have revealed that both ozonation and AOPs are very effective in achieving excellent removals (˃99%) of EDCs from different water matrices, having achieved good results in the removal of persistent EDCs organic microcontaminants. However, inadequate capacity to mineralize the organic contaminants, persistent detection of radical scavengers in the effluent restricting the attack of the radicals to the contaminants, significant interference of natural organic matter with ozone decomposition rate, formation of toxic by-products, oxidation intermediates, solid by-products (mostly unknown), and high operating cost are some of the major drawbacks that can reduce their wide application at full scale.

### 3.6. Removal of Endocrine-Disrupting Compounds via Adsorption Process

Adsorption is one of the most effective methods for treating wastewater, and it essentially depends on the availability of active sites on the sorbent, surface chemistry, and also the chemical (sorbate pKa, basicity or acidity of the sorbent, etc.) and physical properties (such as the sorbate molecular size, sorbent pore density, contact area, etc.), and the specific interactions between adsorbent–adsorbate [102,155]. However, the adsorbent and adsorbate may have distinct properties based on their constituents, and this is the key determinant of the type of adsorption [156,157]. Generally, the adsorption process may be considered as physisorption and chemisorption [102,158]. The processes may occur in different interfaces such as solid–liquid and/or solid–gas in the presence of interactive forces between the surfaces [159,160]. The physical interaction between the adsorbed compounds and the solid surface due to weak van der Waals force of attraction results in the reversible process called physisorption. The fundamental interaction of permanent and temporary electric dipoles generates the van der Waal forces. Principally, the adsorbate is at a distance from the interacting active plane surface but entrapped due to the binding energy, and this allows multiple layers or a single layer of adsorption [158].

As a result of the weak binding energy, a lower temperature is required for the desorption process. The activation energy usually ranges between 20 and 40 kJ, which implies that the tendency of the active sorbent in an aqueous medium to dissolve is high [161]. This could subvert the overall adsorption capacity, though the mechanism correlates with the treatment factors, particularly pH, dosage, particle size, temperature, contact time, and agitation speed [158]. Moreover, most adsorbents have an excellent potential for regenerating adsorption capacity and the release of quality and safe effluent suitable for discharge [162]. 

Figure 3a,b depict a schematic physisorption and chemisorption mechanism, respectively. In the chemisorption mechanism, the chemical bonding results in the breakage and formation of a new bond between the active plane surface of the sorbent and adsorbate [158]. This signifies that higher adsorption energy and temperature is required, which is usually in the range between 200 and 400 kJ/mol [161]. Distinctly, single layer adsorbate occurs in chemisorption, and the mechanism is influenced by the aforementioned treatment factors [163].

Adsorption technique for the removal of EDCs from various water sources using AC has received extensive efforts, which have yielded considerable progress in the last two decades [164,165,166]. For example, Temmink et al. [139] and Kovalova et al. [25] in their separate studies reported excellent EDCs removal using PAC, ranging between 84% and 99% under different operating adsorption conditions. The summary of findings regarding the removal of EDCs via adsorption is presented in Table 6.

Han et al. [70] used 0.2 g/L of nonporous polyamide adsorbent with varying pH (4.8–9.1) to eliminate 17α-ethinyl estradiol (EE2) from the water. An adsorption capacity of 25.4 mg/g was recorded, regardless of variation in water chemistry parameters and pH. Though the experiment was a molecule-level investigation, Zhang et al. [170] reported a maximum adsorption capacity of 476 mgg^−1^ (7.8 µmol(m^2^)^−1^) and adsorption equilibrium at pH 6.5 within 10 h during the adsorption of cholorotetracycline (CTC) on Fe_3_O_4_ magnetite nanoparticles. These results showed that the adsorption of CTC was not affected by ions strength and low concentrations of coexisting Ca^2+^ and mg^2+^. Similarly, Zhang et al. [171] proved that a magnetic hyper-crosslinked microsphere (NAND-−1) could effectively extract EDC pollutants from water samples. This was demonstrated by applying 50 mg/L of NAND-1 to a 5 L solution at pH 2, which reached equilibrium in a short amount of time (30 min). At optimal conditions, 91.7% to 99.4% recoveries were attained for the EDCs in a shorter period of 30 min. The authors established that the NAND-1 could be reused ten times and still achieve recoveries of the target EDCs higher than 86%. Moreover, Grover et al. [172] reported that full-scale GAC removes steroidal estrogens from sewage water excellently, but a lower performance was recorded with pharmaceutical compounds in wastewater. Conversely, Yang et al. [66] indicated that GAC removal capacities differed due to the nature of the EDC contaminants. This is because almost 100% and 75% removal of diclofenac and carbamazepine were obtained, respectively, compared to 45% for the removal of caffeine. 

Solak et al. [41] performed a batch adsorption experiment using highly crosslinked polymer adsorbent (0.2–1.2 g/L) and activated carbon (0.05–0.2 g/L) for between 5.7 and 24.2 min retention time. The result of the study revealed that polymer adsorbents demonstrated good efficiency for eliminating EDCs from water. Authors suggested that their application in biologically treated sewage for the removal of EDCs is a feasible alternative. However, the authors discovered that the components of antibiotics resistance of the EDCs were detected in the treated effluent, as well as significant deterioration of the EDCs removal rate when treated urban sewage was considered as the water matrix. Alizadeh Fard et al. [169] explored the application of 0.1 mg dosage of PVP-coated magnetite nanoparticles to eliminate some selected EDCs from water. The result of the adsorption showed that over 98% and 95% of bisphenol A (BPA) and ketoproven were successfully removed within 15 min, respectively. Recently, Adebayo et al. [167] applied similar adsorbents (0.2 g) and achieved optimum EDCs removal at 60 min contact time. These results revealed that the process was successful and favourable at the optimum time of 60 min at pH 6. Similarly, Wang et al. [173] conducted a comparative study by comparing the performance of magnetite nanoparticles sorbent with AC in removing EDCs from water. The former sorbent recorded an upturn in the removal of PFOA and PFOS with 92–95% and 94–97% within 2 min, respectively, as compared to AC.

#### Factors Influencing the Removal of EDCs during Adsorption Process

To attain a higher removal rate of EDCs, there is a critical need to clearly understand the factors that determine the removal of EDCs in an adsorption experiment. These factors include the solution pH, nature and dose of adsorbent, surface area, temperature, initial concentration of adsorbate, contact time, agitation speed, cationic exchange capacity (CEC), and interfering substances [158,162,174]. These parameters could undermine the adsorption technique and control the sorption rate of solutes towards the sorbent.

The pH of the solution containing the adsorbate and adsorbent is a major and influential parameter, performing a pivotal role in the process of biosorption. Particularly, sorption is predominantly dependent on the pH [175]. This is because pH has a serious impact on the active site of biosorbent surface (ionization of functional group), sorbent surface charge density, and chemical substances of sorbate in water solutions [176]. Regardless, the solution pH has a potent influence on the vast majority of bio-sorbents species. Most H^+^ ions exist in strongly acidic conditions, whereas OH^-^ are superfluous in the alkaline area, and thus, affect the sorption process performance. For example, sorption of negatively charged contaminants (metal anions) is highly beneficial in acidic conditions, as a result of the protonation of a binding functional group without competing with OH^-^, resulting in a better performance of biosorption process. Additionally, moderate pH could substantially affect sorbate solution chemistry (speciation). The level at which the electric charge density on the surface is equal to zero on the airfoil of the adsorbent media is described as the point of zero charges (pzc). It is the point (pH value) where the anions and cations on the sorbent airfoil are equal, and it is illustrated in terms of the solution’s absorption. When the solution pH is below pH_pzc_, more protons are donated by the acid medium than the hydroxide group. Hence, the sorbent surface is positively charged and facilitates the subsequent adsorption of anions. In contrast, when the solution’s pH is above pH_pzc_, the sorbent surface is negatively charged, thus enhancing the sorption of cations from the solution [162].

The selection of adsorbent is defined by its ability to reduce the priority pollutants in wastewater. Adsorbents that readily adsorb non-polar solutes are hydrophobic, while polar solutes are readily sorbed towards the hydrophilic surface [177]. The properties of adsorbent are determined by the angle between the liquid drops in contact with the interface between the adsorbent and adsorbate. Hydrophobic adsorbents have a contact angle above 90°, while a surface that readily dissolves in aqueous solution for the adsorption of pollutants is known to have a contact angle below 90° [175].

Comparatively, being a surface phenomenon, the adsorption rate is proportional to the effective surface area (section of the entire surface available for sorption) [178]. Hence, the more porous and more finely divided is a solid, the higher is the volume of sorption achieved for a unit weight of a solid sorbent. The significant contributions to the surface area are positioned in the molecular proportional pores [24].

The cationic exchange capacity (CEC) is the volume of a negatively charged site accessible on the surface of the adsorbent capable of carrying positively charged ions, otherwise known as functional groups (including Ca^2+^, Mg^2+^, and K^+^), through the electrostatic interaction. The cations retained as a result of the electrostatic force are easily exchangeable with the cations in the effluent. Thus, the biosorbent with increased CEC can stimulate more non-ionic interaction in solution than those with relatively low CEC [159].

Notably, adsorbent dose and contact time are significant variables that determine the rate and capacity of the adsorption process [179]. Either longer contact time or higher dose can probably result in higher EDCs removal; thus, providing the right mix of each will result in good working conditions in a large-scale facility [24]. Similarly, Luo et al. [26] opined that contact time is the leading parameter that influences the adsorption capacity during the adsorption process; thus, shorter contact time is responsible for reduced adsorption capacity.

The temperature of the solution is another critical factor, since adsorption reactions are exothermic; temperature drops could result in increased adsorption capacity [174,180].

The physicochemical properties of the adsorbent and solute solubility could severely undermine the adsorption capacity, rate of sorption, and adsorption equilibrium [24]. Another essential parameter that could influence the elimination of EDCs in water via the adsorption technique is the water-octanol partition coefficient (logK_ow_). This is because hydrophobic pollutants with logK_ow_ greater than 4 (logK_ow_ > 4) have higher adsorption capacities [176].

It is pertinent to note that the adsorption process could also be influenced by the concentration of inorganic and organic molecules. This implies that the combination of several compounds that are normally available in water has a high influence on the adsorption process. These compounds could jointly improve adsorption, operate independently, or negatively impact one another. Principally, natural organic matter (NOM) has a devastating impact on the adsorption of EDCs in both surface and sewage water [181].

Comparatively, activated carbon has become the most consistently utilized sorbent in treating wastewater globally, as highlighted earlier. Studies regarding the elimination of EDC pollutants from aqueous solutions using adsorption test are accomplished by applying activated carbon to implement the technology in full-scale studies [24]. Specifically, the efficiency of PAC in the elimination of EDCs relies on the contact time, pH, surface charge, PAC dose, molecular structure, and characteristics of the target compound and water chemistry [182]. In addition, Bolong et al. [36] stated that the elimination of bio-pollutants from wastewater increasingly depends upon competition for sorption sites, particle–contaminant interactions, and pore blocking (solid particles), which affect the adsorption capacity of activated carbon. As regards GAC, Snyder et al. [179] asserted that GAC tends to perform unsatisfactorily in significantly contaminated wastewater and is less effective in the removal of EDCs with a lower water partition coefficient (K_ow_), which implies that steam-treated GAC could probably be used to overcome the flaw of GAC as a result of its higher adsorption capacity.

Additionally, routine recovery of GAC, carbon type, surface charge of compounds, pore size/shape, volumes of activated carbons, and operation year are pivotal to a sustained breakthrough in the performance of GAC in eliminating EDCs [183,184].

To recap, the application of adsorption-based processes as green technology has demonstrated high potential, and they could be effective solutions over other treatment options for the elimination of EDCs from water, as they are physical treatment processes that do not generate unsatisfactory by-products. Besides, adsorption techniques are outstanding compared to other sewage treatment approaches as regards ease of configuration and application, insensitivity to noxious compounds, cost-effectiveness, and comparatively small footprint. Furthermore, it has been proved that GAC and PAC emerged as desirable techniques for the elimination of EDCs from wastewater and water matrices. Generally, adequate removal is probably feasible especially with the non-polar compounds (K_ow_˃2), with corresponding pore diameter and technical requirements [185]. Though the use of activated carbon is restricted owing to its high cost, copresence of natural organic matter vying for active sites leads to blockage of pores, making it less effective for removing low concentrations of EDCs in wastewater. Besides, the production of PAC is energy intensive, the PAC regeneration process may result in carbon loss, and the recycled product could have a slight decline in adsorption potential. The utilization of the adsorption method for the remediation of EDCs in water has received serious interest among scholars. However, the quest for affordable sorbents with contaminant-binding capacities has improved in recent years. Materials extracted from agricultural, industrial, and natural waste materials could be utilized as sorbent materials for wastewater treatment. Besides, these low-cost adsorbents would provide the adsorption method a practicable solution for water containing EDC contaminants. The choice of a suitable sorbent is a critical issue to attain higher removal of pollutants based on the adsorbate and adsorbent properties. Adsorption technique can be incorporated or combined with other methods as a synergistic technology for effective remediation of recalcitrant EDCs in water.

In the last few years, the need for alternative and efficient adsorbent with high regeneration capacity has been extensively studied to address the shortcomings of the conventional AC during the sorption process, as discussed earlier.

Intensive studies and breakthroughs in the field of magnetic nanoparticles have been reported in recent years. These materials often have distinctive properties, such as structural, chemical, magnetic, and electrical properties, which are designed to allow for a multitude of novel applications in the fields of nano-sensors, chemical and biochemical separation, and environmental management [186].

In particular, magnetic nanoparticles (Fe_3_O_4_) have received increasing attention in the field of pollutants adsorption and pollution control due to their unique properties, such as higher removal efficiency, satisfactory stability, ease of preparation, large specific surface area, satisfactory operation, a wide range of binding sites, and strong magnetic properties, which could result in significant adsorption capacities, increased rate of pollutants removal, ease and rapid extraction of sorbent from solvent via a magnetic field, and facile recycling [182,187]. The pollutants are often easily extracted from nanoparticles after magnetic separation by de-sorbent agents, and recycled magnetite nanoparticles (MNPs) could be reused [188]. The Fe_3_O_4_ nanoparticles have a high potential to eliminate various organic contaminants, including EDCs, due to their high adsorptive capacity and rapid adsorption capacity rate for these contaminants [167,173]. Hence, the application of nano-sorbent, particularly Fe_3_O_4_ nanoparticles, will further make adsorption an affordable and efficient approach in the elimination of EDCs from water.

### 3.7. Removal of EDCs Using Hybrid Treatment Processes

In the quest for successful elimination of recalcitrant EDC contaminants, the possibility of combining different treatment processes (hybrid) to effectively eliminate these recalcitrant EDC micropollutants from water has been attempted by many researchers (Figure 4). Table 7 presents a summary of the existing literature on a hybrid process to eliminate EDCs from water.

To enhance the efficiency of the conventional treatment technique, Sahar et al. [197] explored the incorporation of reverse osmosis (RO) with MBR to remove EDCs from municipal wastewater. The result revealed that over 95% removal of diclofenac and almost 100% removal of organic pollutants were achieved, respectively. Nevertheless, the major drawback of this study was the outrageous running cost of the RO system due to the frequent need for de-fouling coupled with energy demand. Limited ozone dose of 2.5 mg/L with permeate flux of 100 L/m^2^h also showed significant removal efficiencies (64–100%) for all the target contaminants when the ozonation process was combined with ultrafiltration membrane to remove EDCs from drinking water [189]. Authors found that ozonation in the membrane tank enhances the performance of multiple-contaminant removal when integrated with a UF membrane, thereby focusing on mitigating the fouling rate. The findings of this study have to be seen in the light of some drawbacks associated with the ozonation process due to potential toxicity of the oxidation by-products of EDCs and high energy demand for ozone generation.

Similarly, Si et al. [193] investigated the performance of combined ozonation and UF processes for the elimination of EDC pollutants in secondary sewage. The findings of the investigation showed that the combination of ozonation (O_3_) and UF showed an excellent result, with almost 100% EDCs removal. Yet, the high cost of ozone production, generation of bromate ions, and organic and solid by-products are the major constraints of this approach. Borikar et al. [194] evaluated and compared conventional and advanced oxidation techniques for the elimination of EDCs and pharmaceutical chemicals from water. Their research findings revealed that ozone/H_2_O_2_, UV/H_2_O_2_, and conventional treatment significantly removed EDCs (97%) from water. In addition, conventional treatment with high UV/H_2_O_2_ also demonstrated effective removal of 92% ± 7%. Interestingly, triclosan, bisphenol A (BPA), and diclofenac were completely eliminated. However, the conventional treatment poorly removed selected EDCs and pharmaceutical compounds. The combination of ozonation and ultrafiltration showed a significant removal of almost 100% for almost all the target EDC contaminants. The authors stated that the combination of ozone and UF provides an efficient technique to regulate the concentration and contamination of EDCs in secondary wastewater. However, the major drawbacks in this study are the potential bromate formation and solid by-products from the ozonation process, as well as the huge cost of ozone generation. Hu et al. [38] studied the elimination of EDC contaminants from secondary sewage by combining the UF membrane and the ozonation process. Results of the study showed that effective removal of EDCs (average 75%) and estrogenicity (84%) were achieved, respectively. The authors suggested that efficient removal of EDCs from secondary effluent with weakened membrane fouling could be achieved by combining ozonation and the UF membrane process. However, this study is subject to several limitations, such as longer operating time (52 days), the possible generation of bromate ions and organic and other solid by-products during the ozonation process, and the high cost of ozone generation (most especially in a pilot-scale system), and the efficiencies on most operational days was less than 70%, which was too low compared to RO and NF membrane.

In addition, Acero et al. [190] used a hybrid process of adsorption, coagulation, and ultrafiltration to regulate the EDCs concentration in water. They stated that coupling UF with activated carbon is capable of producing reusable water free of EDC micropollutants, particularly if a hydrophilic membrane is used. Similarly, Li et al. [191] studied the removal of 17α-ethynylestradiol (EE2) from water by applying a combination of activated carbon (AC) adsorption and a PAC/UF membrane process. Appreciable removal efficiency of 7.0–80.0% was reported. The major drawbacks of this study were the high cost and regeneration problems associated with PAC; copresence of natural organic matter which may compete for sorption sites, thereby leading to blocked pores; and technical challenges attributed to the separation of PAC from the treated water.

Silva et al. [45] reported the elimination of 17β-estradiol (E2), 17α-ethynylestradiol (EE2), and estriol (E3) from treated domestic wastewater using advanced oxidative processes and a membrane filtration system. Considerable removal of EE2 (almost 70%) was achieved. Further optimization using an experimental design with a higher UV dose (122.4 kJm^−2^) and low H_2_O_2_ concentration (4 mg/L) yielded significant removal efficiencies of 91% for E3 and 100% for E2 and EE2, respectively. This process shows very strong potential for the removal of target compounds, but its application might be restricted due to the expensive nature of the advanced oxidation process, potential obstruction of UV light penetration by turbidity, longer filtration time, and potential bromated by-products. In a pilot-scale experiment by James et al. [196], the removal of EDCs from secondary urban sewage was studied using an advanced oxidation process (AOP) relying on UV irradiation in combination with hydrogen peroxide incorporated with MF and RO. AOP satisfactorily reduced EDCs levels and showed a higher removal of more than 90% for all waters. Importantly, results of the study revealed that significant removals of estrogenic compounds up to 99% was achieved. Despite this higher removal, some shortcomings of this study include high energy demand, existence of some practical limitations, inevitable cost penalty, and potential bromated by-product compounds, such as nitrate, which can interfere with the absorbance of UV light.

Recently, Ferreiro et al. [8] studied EDCs removal from effluent using an existing biological treatment combined with ultrafiltration (continuous-UF) as advanced treatment. Results revealed that some of the EDCs showed excellent removal, with an upturn of 99.5%. Though the removal performance of this hybrid system is grossly inconsistent, and this might be due to the various EDCs’ biodegradability and concentrations. Similarly, the frequent fouling and longer retention time (high HRT) undermine the merit of this treatment method. 

Overall, the hybrid processes reported a remarkable removal performance, ranging from 64% to 100% for remediating different water sources. Among the hybrid processes, combined ozonation and advanced oxidation processes with the UF membrane showed an upturn performance in most cases. Moreover, they are the most widely used treatment techniques in combination with other treatment methods, as regards hybrid processes for the removal of EDCs in recent years. It is also important to note that some of the problems associated with AOP and ozonation processes in terms of the production of noxious by-products (namely bromate ion), oxidation intermediates, and the consequently high operating cost due to membrane fouling.

## 4. Conclusions and Future Prospects

The continuous persistence of ubiquitous endocrine-disrupting compounds in receiving waters, from ng/L to µg/L, has become a serious threat to the global waters and ecosystem due to their direct and indirect influence on the health of humans and the environment. Occurrences, sources, various detrimental impacts of EDCs on humans, aquatic, wildlife, ecosystem, and recent advances in treatment techniques have been sufficiently reviewed. Effluent released from wastewater treatment facilities have been reported and identified as the main root of several EDC contaminants, owing to their diverse properties (such as biodegradability and hydrophobicity) even at low concentrations. This is inextricably linked to the failure of the current conventional wastewater treatment facilities to efficiently eliminate recalcitrant and refractory emerging EDC contaminants from wastewater effluent, due to the complexity and persistence of the compounds, thereby posing a critical challenge for water management industries and consumers. Lack of standard discharge limits and guidelines for EDCs and other new compounds with regards to water treatment industries has further propelled the need to continuously monitor the proportions of these persistent contaminants, as it is anticipated that their occurrences in the effluent from the conventional treatment systems will generate more severe hazards in the near future as a result of rapid increasing industrial and technological breakthroughs and rising demand of the industrial sector. In this regard, this paper suggests stringent regional or international discharge standards for microcontaminants into the environment to curb the menace of persistent occurrence and regulate the concentration of these emerging contaminants in the environmental compartments.

Consequently, recent research trends in the techniques of eliminating endocrine-disrupting compounds from various water sources have been discussed and applied in various regions of the globe. Although these technologies have proven to be promising alternatives for EDC removal, most of the methods are faced with associated drawbacks in tackling these persistent EDCs contaminants, resulting to difficulties in the provision of secure and safe water supplies, and therefore are not suitable for large-scale treatment of such persistent pollutants. Besides, these treatment technologies are generally exorbitant, and their application could lead to other challenges, particularly the generation of noxious by-products, toxic sludge, concentrated residues, complex procedures, and high operation and maintenance costs, which reduces their applicability.

It is worth mentioning that this review article has offered a unique focus on the various inherent drawbacks of various wastewater treatment technologies, which have not been sufficiently discussed in the literature. Hence, an effective treatment process for the elimination of EDC contaminants is highly requisite. In this context, the development of sustainable, low-cost, and environmentally friendly hybrid systems, such as adsorption-nano-composite membrane, is a promising and novel approach. A viable alternative to eliminate recalcitrant environmental EDC pollutants from entering our water resources is urgently required, and must be a physically integrated treatment technique with unique properties, such as zero yield of undesirable by-products, facile operational process, ease of configuration and application, little or zero addition of chemicals, ability to remove EDCs without phase change, insensitivity to noxious compounds, cost-effectiveness, and comparatively small footprint.

## 5. Future Prospects for the Removal of EDCs via Contemporary Techniques

Further studies are required to develop an eco-friendly integrated wastewater treatment system with excellent working conditions, such as adsorbent dosage, contact time, membrane flux, transmembrane pressure (TMP), and temperature, among others. The integrated system includes the ideal adsorption process using nano-sorbent prior to membrane filtration in a single and one-term system. Furthermore, to be able to develop such a treatment system, a good understanding of the choice of different nano-sorbent- and polymer-based adsorbents with specific adsorption capacity is, therefore, a critical issue to ensure optimal elimination of EDC contaminants from various water sources, and the associated removal mechanisms are essential. Indispensably, the combination of adsorption and membrane system will curtail susceptibility to fouling during the removal of EDC contaminants due to the synergetic effect of the integrated technology. The integrated technology could also enhance filtration efficiency, lead to excellent EDC contaminants removal, and result in a remarkable decrease in overall treatment cost.

## Figures and Tables

**Figure 1 polymers-13-03229-f001:**
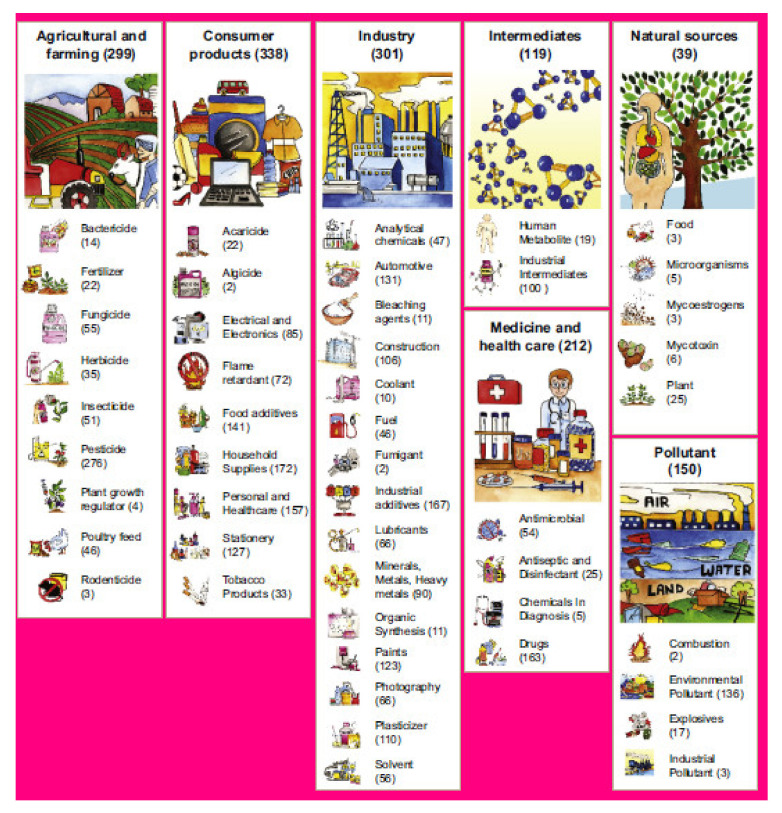
Classification of EDC pollutants [28].

**Figure 2 polymers-13-03229-f002:**
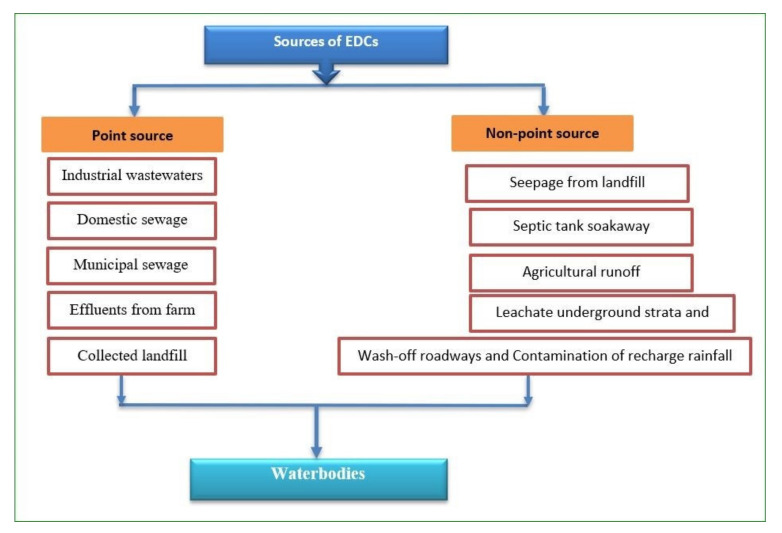
Representative sources of endocrine-disrupting compounds (EDCs) in the environmental matrices [37].

**Figure 3 polymers-13-03229-f003:**
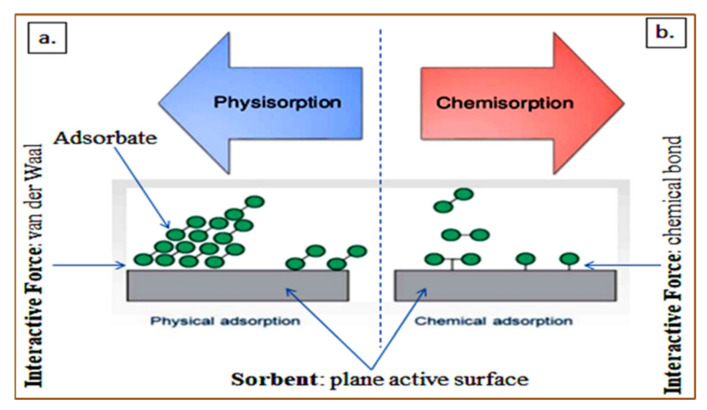
Adsorption mechanisms of (**a**) physisorption and (**b**) chemisorption. (https://www.slideshare.net/jaskiratkaur28/adsorption-isotherms-81552835) accessed 20 June 2021.

**Figure 4 polymers-13-03229-f004:**
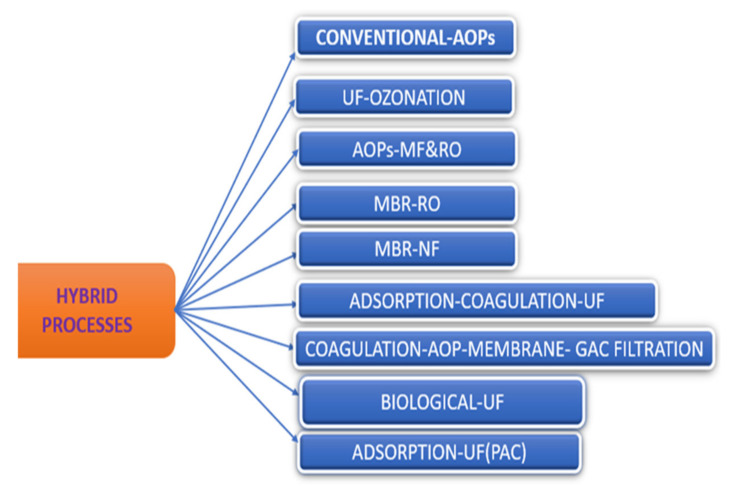
An overview of hybrid systems for the elimination of EDCs pollutants from various water sources.

**Table 1 polymers-13-03229-t001:** Published information on different concentrations and deleterious effects of endocrine-disrupting compounds from different routes.

Matrices Type	Major Pollutants	Corresponding Proportions (ng/L)	Major Effects	References
Secondary effluent from the municipal wastewater treatment plant	E1, E2, E3, EE2, BPA	21.2; 162.7; 2.4; 138.3; 23.9	Series of health problems and present potential risks to the ecosystem.	[39,52]
Treated wastewater	2,4- dichloro phenol 2,3,4- trichloro phenol E1, E2, EE2	1.00; 0.80; 1.20; 1.20; 1.20	Effects on human health, wildlife, and fisheries (or their progenies) by interaction with the endocrine system.	[41]
Biotreated sewage	E2, EE2, E3	24.46; 34.18; 826.68	Interruption of normal function of the endocrine system of wildlife by inhibiting, imitating, or acting like natural hormones and decrease of testosterone level. It heightened shoaling and anxiety behavior.	[46,54]
Municipal wastewater, bottle and ultrapure water	BPA	50,000	Increased incidence of cancer, decreasing reproductive fitness of men, and threat to aquatic organisms and humans.	[43]
Synthetic municipal wastewater	BPA	16,300	Interruption of reproductive mating behaviors of fish species. Stimulation of breast cancer cells.	[55,56]
Ultra-pure water (Milli-Q water)	E1, E2, EE2	13,330; 8550; 9170	Interference with the body’s endocrine system by influencing the synthesis, release, transport, metabolism, and excretion of hormones in the body. Influence on the thyroid, adrenal gland functions, and developmental dysfunctions. It causes reproductive, immune, neurological diseases.	[57,58]
Secondary wastewater effluent	E1, E2, EE2, DES, TCS	NA	Impacts on water quality and potential hazards to aquatic organisms and public health.	[59]
Urban wastewater effluents	E1, BPA, E2, EE2, TCS, TST, SAL	NA	Biomagnification and bioaccumulation through the food supply chain. Elevated trophic-level species in humans and the ecosystem via food intake	[60]
Surface water	E1, E2, E3, EE2	1.40–5.74; 1.10–5.39; 2.15–5.20; 11.70–14.00	Physical effect on human health and water environment.	[61,62]
Wastewater from the wastewater treatment plant	Diltiazem, CBZ, Acetaminophen, E1, Progesterone	14.5; 93; 860; 185; 20	Disruption of the delicate balance of the endocrine system of mammals. Gender shifts and reduced fecundity in fish.	[42]
Wastewater sample from the wastewater treatment facility	E1, E2, EE2, E3, BPA, NP	169; 21.3; 125.9; 41.6; 1511; 5002	Disruption of the normal hormone functions and physiological status in human beings and animals.	[63]
Biologically treated wastewater	BPA	500,000 ± 0.19	An environmental toxic substance with relatively high biological activity.	[53]
Marine sediment	Testosterone, Progesterone, E1, E2, Dexamethasone, Primidone, Propranolol, Atorvastatin, SFZ, Diclofenac, DES, Nitrofurazone, BPA	<0.015–0.094; NA; 0.014–0.038 NA; NA; 0.040–0.144; 0.020–0.097; 0.250–0.275; NA; NA; 0.093–0.228; NA; 0.072–0.389	They are mimicking and blocking the endocrine system in mammals with acute cancer, irregular reproductive development, and metabolic malfunctions (obesity and diabetes). Biomagnification and bioaccumulation in the marine ecosystem, associated with potentially skewed sex ratios, intersex, and weak gonadal growth and viability.	[47]
Prepared Stock solution and	E2	1,000,000	Impacts on wildlife and human health.	[64]
Malaysia tropical waters	EE2, Levonorgestrel, Norethindrone, Cyproterone acetate	1898; 213; 11,336; 262	Inhibition of testicular growth and inducing intersex changes. Inhibition of reproduction in adult fathead minnows. Reproductive impairment and variation in sex ratio.	[48] ([45,49])
Groundwater	E1, E2	55.1; 56.1	Interference with the functions of the endocrine system.	[65,66]
Surface sediment samples (Anzali wetland)	4- NP, OP, BPA,	29; 4.3; 7 (µg/g dry weight)	Mimicking actions of endogenous estrogens, thereby causing reproductive disorders. Feminization and carcinogenesis in numerous organisms.	[67]
River water	BPA	215	NA	[68]
Fish muscle	BPA, 4- NP, OP	0.023–0.322; 0.124; 0.023 (ng/g)	NA	[69]
Estuarine water	Testosterone, Progesterone, Dexamethasone, Primidone, Propranolol, Caffeine, SFZ, Diclofenac, Chloramphenicol, Diazinon, E1, E2, EE2, BPA	0.51–2.30; <0.41–0.46; <1.00–1.51; NA; 0.25–0.34; 0.13–0.33; NA; 0.47–79.72; <0.05–0.09; NA; <0.56–1.92; 5.28–31.11; <0.30–7.69; 0.19–0.47.	Mimicking actions of hormones from fetal to adult stage of development of a living organism. Negative impacts on the hormonal systems of organisms.	[47]
Aqueous solution	EE2	NA	Ultra-high estrogenicity.	[70]

E1, estrone; CBZ, carbamazepine; EE2, 17α-ethinyl estradiol; DES, diethylstilbestrol; E3, estriol; OP1EO, octyl phenol monoethoxylate; BPA, bisphenol A; NP, nonylphenol; E2-3S, 17β-estradiol glucuronide; OP, octyl-phenol; SFZ, sulfamethoxazole; E2, 17β-estradiol; NP2EO, nonylphenol diethoxylate; DMP, dimethyl phthalate; NP1EO, nonylphenol monoethoxylate; TCS, triclosan; NP2EC, nonyl phenoxy acetic acid; DEP, diethyl phthalate; OP2EO, octyl phenol di ethoxylate; E1-3G, estrone-3-gluclonide; 4 tOP, 4-tert-octyl-phenol; E1-3S, estrone-3-sulfate;.

**Table 2 polymers-13-03229-t002:** Treatment by conventional processes.

Major Contaminants/Sources	Treatment Process	Treatment Factor	Brief Procedure	Major Findings	Limitations	References
TCS, NP2EO, IBF, DCF, TCS, BPA, KFN, NP, NP1EO, NPX/wastewater, and sewage sludge samples	Conventional treatment (mesophilic anaerobic sludge digestion)	(HRT: 9 h; SRT: 8 d), (SRT: 17 d). (HRT: 23 h, SRT: 18 d).	Wastewater samples collection. Sewage sludge samples were homogenized extractions of wastewater samples.	The removal efficiency of DCF and IBF ranged between 39% and 100%, IBF and NPX were ˃80%.	Higher proportions of NP in digested sludge. Detection of TCS and NP in treated wastewater. Too many modular units.	[74]
E1, E2, E3, EE2, BPA, and 4-NP/rural wastewater effluent	Activated sludge. Micro-power biofilm reactor. Constructed wetland. Stabilization pond.	Temp: 30 and 70 °C; HRT = 12–24; 24–120; 24–240; 10–16.	Biological contact oxidation. Subsurface flow. Facultative pond. Anoxic oxidation.	Percent removal of target EDCs > 70% in summer.	Unstable performance of decentralized processes. Pronounced impacts of effluent discharged on the quality of receiving water. Too many modular units.	[63]
BPA, E1, E2, E3, EE2, and DES/effluent from a wastewater treatment plant	WWTP activated sludge treatment processes. Oxidation ditch reversed anaerobic and sequential batch reactor SBR.	HRT: 7.6–35.31 h SRT: 5.8–31.9 days		73.7% of BPA was removed. High removal rates of EDCs (i.e., > 85%).	Some concentrations of EDCs were found in the effluents and can pose potential risks to ecosystems and human health. Longer HRT and SRT.	[75]
59 EDC contaminants/ wastewater effluents	Fluidized powdered activated carbon (PAC) pilot (WWTP configuration).	SRT: 5–7 days; bed depth: 1–3 m. hydraulic velocity: 6–12 m/h; contact time: 10–20 min.	Pre-primary and biological treatments. Pre-treatment (screening). Biofiltration system, Micropollutant analyses.	Removal of parabens and pesticides ranged between 50% and 95%, paracetamol, IBF, sulfamethoxazole 60–80%	Artificial sweeteners (1000–10,000 ng/L), BPA and NP (100–1000 ng/L) were detected in the effluent.	[21]

TCS, triclosan; E1, estrone; DES, diethylstilbestrol; BPA, bisphenol A; IBF, ibuprofen; NP, nonylphenol; E2, 17β- estradiol; NPX, naproxen; OP, octyl-phenol; EE2, 17α-ethinyl estradiol; DCF, diclofenac; NP1EO, nonylphenol monoethoxylate; E3, estriol; 4 NP, 4-nonyl-phenol; KFN, ketoprofen; HRT, hydraulic retention time; WWTP, wastewater treatment plant; SRT, solid retention time; NP2EO, nonylphenol diethoxylate.

**Table 3 polymers-13-03229-t003:** Removals of some EDCs during photocatalytic and enzymatic degradation processes.

Major Contaminants/Source	Treatment Process	Treatment Factor	Brief Procedure	Major Findings	Limitations	References
EE2/ ultra-pure water and treated wastewater	Photocatalytic degradation using ZnO under simulated solar radiation	EE2 conc: 100–500 µg/L, photon flux: 4.93 × 10^−7^–5.8 × 10^−7^ Einstein L^−1^ S^−1^; ZnO conc: 50–500 mg/L, treatment time: 2–10 min.	Spiking of water matrix was spiked with EE2, photocatalysis of the solution. Periodic sampling and centrifugation.	Rapid EE2 degradation occurred via first-order kinetics.	Detection of EE2 in the effluent. Retardation of EE2 degradation by organic and inorganic matter.	[86]
BPA/municipal WWTP, bottled water, ultra-pure water	Solar photocatalytic degradation	pH: 6.1, catalyst: 81.3–339.2 mg, TiO_2_ loading: 0, 81.3, 101.8, 152.3, 339.2 mg, ZnO loading: 0.5–6.8 mg/cm^2^, H2O2: (25–100 mg/L), BPA initial conc.: 50–200 µg/L, treatment time: 0–90 min.	The incident radiation intensity was measured econometrically. The water matrix was spiked with the organic substance with the addition of the ZnO /TiO_2_ catalytic plate. Periodic sampling and analysis.	Increasing the number of immobilized catalysts enhances BPA conversion.	Partial inhibition of BPA degradation due to the presence of EE2. Weak degradation in wastewater.	[43]
E1, E2, EE2, E3, NP, BPA/artificial, and real wastewater	Enzymatic degradation using fungal laccases	pH: 1–1.5 Temperature: −20 °C Contact time: 2, 6, 24 h.	Constant shaking as laccase uses molecular oxygen for oxidizing substrates. Acidification of enzymatic reaction at each time interval (2, 6, and 24 h). Complete inactivation of the laccase activity. Extraction via solid-phase extraction (SPE) for chemical analysis.	Immobilized laccase removed EDCs (83% for T. Versicolor and 87% of M. thermophile), 99% removal after 24 h. Removal rates for estrogenic = 82% after 24 h.	Formation of toxic by-products.	[87]
BPA	Fungal laccases degradation using oxidative enzyme	(1): 25 μM of each molecule, pH 5.0 (50 mM sodium citrate buffer), 1.5 U/mL laccase, (2): 100 μM BPA, pH 5.0 (50 mM sodium citrate buffer), 25 °C, and 1.5 U/mL laccase, reaction time: 1 h.	Addition of methanol and Tween to the solution. Incubation of each EDC. Addition of hydrochloric acid (HCl) to the reaction mixture and centrifugation at a specific time interval at room temperature. Analysis of supernatant and BPA degradation.	BPA was oxidized under all conditions tested.	Complex procedure.	[88]
2-chlorophenol and SMZ/municipal wastewater	Laccase degradation	pH 7, initial SMX at 10 μM and ACE at 10 μM. Time (h): 0, 0.25,1, 24.	NA	Excellent removal of SMZ in the absence of mediators in secondary effluent.	Poor removal of sulfamethoxazole in all buffered solutions. Not economically viable.	[89]
BPA, 2,4-dichlorophenol, 4-tert OP, pentachlorophenol, and NP/aquatic plants	Enzymatic degradation	Endogenous H_2_O_2_ concentration in aquatic plants (170–590 μmol/kg-FW)		EDCs were degraded by oxidative enzymes.	Longer treatment period (>100 days). Complex procedure.	[90]
Atrazine (herbicide), phenyl phenol, BPA, and TCS/municipal wastewater	Biosorption and biodegradation.	Feed NaCl concentration (0–15 g/L). Initial MLSS = 16 g/L; HRT = 5 d; mixed liquor pH = 7 ± 0.1; temperature = 35 ± 1 °C.	Feeding the bioreactor, circulation of digested sludge. Mixing of the sludge.	Trimethoprim, carazolol, hydroxyzine, amitriptyline, and linuron, removal rates ≤ 80%. Phenyl phenol removal = 60%.	Relatively low removal rates of phenyl phenol, BPA, and TCS. BPA was poorly removed, from 40% to 20%. Poor removal of atrazine (6.8%).	[91]
DEHP, fluoranthene, AMPA, and E1/ wastewater effluent	Filamentous fungi biodegradation.	pH 5.5, incubation period: 96 h (AT96h), degradation period: 10 days.	Degradation test conducted in mineral medium incubated for 10 days with each fungus.	Fungi degradation of DEHP = 100%, AMPA = 69% with F. solani and T. harzianum.	E1 not degraded by all fungal isolate trials.	[80]

BPA, bisphenol A; E1, estrone; EE2, 17α-ethinyl estradiol; E3, estriol; NP, nonylphenol; E2, 17β- estradiol; OP, octyl-phenol; TCS, triclosan; SMZ, sulfamethoxazole; 4 tOP, 4-tert-octyl-phenol; DEHP, di(2-ethylhexyl) phthalate; AMPA, aminomethylphosphonic acid.

**Table 4 polymers-13-03229-t004:** Removal of EDCs by Membranes.

Major Contaminants/Water	Treatment Process	Operating/Treatment Factor	Brief Procedure	Major Findings	Limitations	References
E1, E2, progesterone, testosterone/purified water	UF membrane	MWCO: 1–100kDa Pressure: 0.5–5 bar Pure water flux (L/m^2^h) 20.8–359.2 Final flux:21.9–288.5 Time: 2–40 min pH: 8	Stirring feed solution at 200 rpm for 16 h. Filtering of purified membrane for 30 min. Measurement of pure water flux. Collection of permeate.	Removal via solute–solute interactions for E1 correspond to higher proportion of organic matter at 25–50 mg/L for 10 kDa (48–52%); 100 kDa (33–38%) membranes.	Poor removals of E1 and hormone contaminants (52% and 38%).	[111]
BPA, CBZ, IBF, and SFZ/drinking water	UF membrane	Operating speed: 50 psi. Flow rate: 0.65 L/min per cell.		Initial partial removal of BPA.	Poor BPA removal using modified PES membranes.	[113]
BPA/drinking water	UF-PS (PS) membrane.	Temp: 25 ± 0.5 °C. pH: 7–13 BPA concentration: 100–500 μg/L. pH: (3.68–10)	Measurement of pure water flux. Filtration of BPA solution.	Higher removal at the initial stage of the filtration.	Lower removal efficiency (20%). Fouling.	[120]
BPA/pure BPA solution	UF membrane	pH: (3–13) MWCO: 100 Da TMP: 0.1 × 10^6^–0.3 × 10^6^ Pa Temp: 20 ± 2 °C BPA conc.: 5 mg/L	The UF membrane was installed and the solution was introduced into the UF cup and followed by magnetic stirring.	Both salt and acidic pH improve the transportation of BPA.	BPA rejection decreased significantly when the BPA molecule was ionized.	[114]
DMP, DEP, DBP, DnOP, DEHP/water	NF membrane	pH: 4–9; pure water flux: 47.5 L/m^2^ h; temperature: 25–45 °C.	Preparation of a feed solution. Measurement of concentrations of PAEs in both the feed and permeate.	Removal efficiencies of 95.4%, 95.1%, and 91.5% were recorded for DEHP, DnOP, and DBP.	Lower adsorption rates. Low rejection of sulfamides.	[131]
BPA/biologically treated wastewater	MF and NF	Suspended solids = 78 ± 12 mg BPA conc.: 0.3 ± 0.14–0.7 ± 0.27 Jv(L/m^2^h) = 6.0–18.6 80 L/(m^2^h) for NF Temp = 21 °C TMP = 0.3 MPa (MF) 0.7 MPa (NF)	Circulation of module with pure water. Determination of pure water infiltration.	Both techniques eliminate BPA. BPA removal efficiency: 61–75% with NF.	Fouling. A decline in permeate flux in MF.	[53]
BPA/model solution	NF and RO membranes	Temperature: 45–50 °C Max pressure: 31–83 bar, pH: 2–11 water permeability: 0.85–14.86 (L/m^2^h bar) Time: 30–360 min		≥98% BPA rejection was achieved with polyamide-based RO membranes.	High energy demand. Too many modular units.	[121]
BPA, E2, E1, E3, EE2/synthetic wastewater	UF membrane	working pressures (25, 30, 50, 75 kPa); temp: 20 ± 2 °C; TOC = 7 mg/L; pH 7.6; conductivity = 1000	Soaking of fresh membrane for 24 h. Removal of impurities. Determination of flux.	EDCs removal rates of (10–76%) were achieved via a fouled membrane.	Poor removal of E3 (10–17%).	[119]
BPA, DMP, DBP, NP, DOP/water	Nano-functionalized membrane using polypropylene (PP) non-woven fabric	Operating pressure: 0.02–0.5 Mpa; pH: 6.5; Temp: 25 °C	The target pollutants were dissolved in deionized water and quantified. The filtration experiment was conducted.	˃80% BPA rejection was recorded after a period of 1.3 s.	Removal of contaminants was attained at higher operating pressure of 0.5 MPa.	[132]
Oxybenzone and BPA/synthetic solution	Nanohybrid (CuSG) blended PES-HF membranes	Filtration time: 120 min; temp: 20 °C; pressure: 1 bar	25 mg/L solution of oxybenzone and 5 mg/L BPA solution were filtrated via the HFM samples and the permeate was analyzed via a UV–visible spectrophotometer	Higher rejection of oxybenzone (98%) and BPA (95%) was recorded. Elevated pure water permeability (528.2 ± 44.6 Ml/m^2^/h/mmHg).	Nil	[133]
BPA/synthetic solution	UF(TFC) immobilized with TiO_2_	Preparation of feed solution. Quantification of the feed and the permeate solution.				[115]
BPA/drinking water	Nanocomposite membrane electrospun PVDF-PVP-MnO_2_	Working pressure: 0.5–2.5 bars; sampling period: 0, 5, 10, 20, and 30 min; temp: 27 °C.	The membrane was fabricated using electrospinning technique and was applied in a filtration system to assess the removal efficiency of BPA. The concentrations of BPA were analyzed using HPLC.	Complete rejection of BPA (100%) was attained for NF2 and NF6 after 30 min.	Nil	[128]
BPA/synthetic solution	Photocatalytic PSF/TiO_2_/Fe-doped composite UF membrane	BPA concentration: 10 mg/L; specific temperature: 140–220 °C, 6–24 h; pressure: 0.1–0.2 MPa.	Preparation of Fe-doped TiO_2_ photocatalysts, synthesis of photocatalytic membranes; assessment of photocatalytic performance	BPA removal rate of 90.78% was recorded.	Nil	[129]
BPA/water	PSF/GO nano-composite membranes	Input pressure: 1–5 bar, operating time: 10–50 min, pH: 3–11, initial BPA concentration: 1–9 mg/L.	Synthesis of GO; preparation of GO/PSF nano-hybrid membranes; BPA concentration was analyzed using a UV–vis spectrophotometer	BPA removal efficiency of 93% was attained.		[130]

E1, estrone; IBF, ibuprofen; E2, 17β-estradiol; SFZ, sulfamethoxazole; EE2, 17α-ethinyl estradiol; PPCPs, pharmaceutical personal care products; BPA, bisphenol A; NP, nonylphenol; TCS, triclosan; E3, estriol; CBZ, carbamazepine; MF, microfiltration; UF, ultrafiltration; NF, nanofiltration; RO, reverse osmosis; PAEs, phthalate acid ester; DMP, dimethyl phthalate; DEP, diethyl phthalate; DBP, dibutyl phthalate; DnOP, Di-n-octyl phthalate; DEHP, diethylhexyl phthalate; NOR, norfloxacin; GO, graphene oxide.

**Table 5 polymers-13-03229-t005:** Removals of EDCs during ozonation and advanced oxidation processes.

Major Contaminants/Sources	Treatment Process	Treatment Factor	Brief Procedure	Major Findings	Limitations	References
Diltiazen, progesterone, BBP, E1, CBZ, acetaminophen/biological sludge	Pulse ozonation experiment	Operating pressure = 5 bar; gas flow rate = 10–140 L/h; MLSS = 2.3–4.2 g/L; ozone period: 6–150 min. Ozone dose (mgO_3_/L): 1.11–18.65; pH = 6.4–7.1	Ozonation of the sludge samples. Continuous aeration. Analysis of the residual EDCs conc. in the samples.	˃99% removal of target EDCs contaminants were achieved after 4 days.	Production of toxic by-products. The high cost of ozone production.	[148]
BPA, E2, and EE2/wastewater	AOP (H_2_O_2_, O_3_, UV, UV/TiO_2_, UV/H_2_O_2_, and UV/O_3_)	NA	NA	The removal rate of pharmaceutical EDCs ≥ 96% during UV/TiO_2_ process.	Poor removal of caffeine. Generation of several oxidation by-products with high toxic potential.	[149]
E2, EE2, BPA/wastewater treatment plant effluent matrix	Degradation by UV light/chlorine	Chlorine conc.: 0.2–2 mg/L; reaction time: 30 min; initial EDC conc.: 100 µg/L; UVC irradiance: 14.79 mW cm^−2^; temp.: 25; pH: 7	Spiking of EDCs in WWTP effluent and ultrapure water. UV/Cl process. Samples collection. Addition of sodium thiosulfate followed by filtration. Disinfection evaluations.	The combination of UVC with chlorine significantly and rapidly degrades EDCs. An upsurge in chlorine concentration yields almost 99% EDCs removal.	Formation of chlorate by-product disinfection. UV light penetration can be obstructed by turbidity.	[150,151]
E1, E2, EE2, DES, TCS, 17α- treubolone, 17 β- treubolone, 19- nortestosterone, AEDb testosterone, methyltestotesterone, 4-OHA, prednisone cortisol, cortison, 19- norethindrone, medroxyprogesterone, BPA, 4-tert-OP, 4- NP, triclocarban, ADD, 17β- boldenone, stanozolol, epi-andosterone, andosterone 5α-dihydrosteterone, preanisolone, dexamethasone, ethynyl testosterone, progesterone/secondary wastewater effluent	Fe (VI) treatment process	Temp. = 23 ± 2 °C micropollutants = 100 µg/L^−1^; Fe (VI) = 10 mgFeL^−1^; pH: 6.88–7.09; Fe (VI) dosage = 0, 2.5, 5, and 10 mgFe L^−1^; DOC. = 5.0 mgCl^−^.	Application of Fe (VI) to secondary effluent. Dosing of solid Fe (VI) in the effluent. Stirring of the solution. Addition of methanol and H_2_SO_4_.	Fe (VI) treatment could achieve both oxidative eliminations of detected EDCs as a tertiary treatment technology.	It failed to react with triclocarban, three androgens. Low ferrate (VI) production rate.	[59,151]
EE2/synthetic secondary effluent	Ozonation	Ozone conc.: 2, 4, 9 mg/L; NOM conc.: 0–80 mg/L; pH: 6–10 O_3_: TOC: 0.2–1.0	Spiking different conc. of ozone into the stock solution. Removal of residual ozone and radicals. Testing of blank controls.	The initial concentration of ozone and natural organic substance adversely affect degradation efficiency. Effective degradation of EE2 by ozonation at pH 6 resulted in higher degradation of EE2.	Generation of toxic by-products. Production of solid by-products. High operating costs.	[152]
BPA/aqueous solution	Microwave-enhanced Mn-Fenton process	BPA initial concentration = 100.0 mg/L; reaction time = 6 min	Addition of BPA solution with Fenton reagents followed by heating. Determination of BPA conc.	BPA removal = 99.7% and total organic carbon (TOC) (53.1%).	Generation of complicated secondary sludge. A narrow range of optimal pH (2.5–4.0).	[153]

E1, estrone; CBZ, carbamazepine; E2, 17β- estradiol; NP, nonylphenol; EE2, 17α-ethinyl estradiol; DES, diethylstilbestrol; BPA, bisphenol A; TCS, triclosan; COD, chemical oxygen demand; AOP, advanced oxidation process; MLSS, mixed liquor suspended solids; NA, not available.

**Table 6 polymers-13-03229-t006:** Removals of some EDCs during the adsorption process.

Major Contaminants/Sources	Treatment Process	Treatment Factor	Brief Procedure	Major Findings	Limitations	References
EE2/water	Adsorption (polyamide adsorbent)	pH: 4.8–9.1; constant dosage of 0.2 g/L; contact time: 24 h; agitation rate: 250 rpm; temp.: 25 °C.	Dilution of EE2 working solutions from EE2 stock solutions. Addition of adsorbent into EE2 aqueous solutions. Agitation of mixed solutions.	Maximum adsorption capacity = 25.4 mg g/L. Adsorption rates ranged between 5.3- and 22.4-fold.	A molecule-level investigation.	[70]
BPA, NP BP3, TCS/aerobically treated greywater	Adsorption (PAC)	29.0 g/70.6 mL bed volume; initial compound proportion: 100–1600 µg/L; dose: 1.25 g/L; contact time: 5 min.	NA	TCS removal = 95%. BPA removal = 99%. NP removal = 84%.	The exorbitant cost of PAC.	[26,139]
TCS, E1, E2, and EE2, clofibric acid, CBZ, clofibrate methyl ester, clofibrate/water, and treated wastewater	Batch adsorption using crosslinked polymer adsorbent and activated carbon	Polymer sorbent dosage: 0.2–1.2 g/L; AC: 0.05–0.2 g/L; retention time: 5.7–24.2 min, temp.: 21 ± 2 °C.	Removal of selected EDCs from ultrapure water. Introduction of polymer adsorbents in solutions of EDCs and agitation.	TCS = 92%, CBZ = 90.5%, E1 = 71.4%, EE2 = 71.3% removals.	Poor contaminants removal using AC when treated municipal wastewater was used.	[41]
BPA/DI water	Batch Adsorption (nano-magnetite)	Adsorption time: 0–120 min; pH: 2–12; adsorbent dose: 0.04–0.22 g; BPA conc: 10–75 ppm; temperature: 30, 35, 40, 45, 50, 55, and 60 °C.	Introduction of 0.1 g of magnetite into different conc. of BPA. Solutions agitation for 45 min at 30 °C. Measurement of residual BPA conc.	Synthetized magnetite offers great potential for the remediation of BPA-contaminated media.	Low adsorption capacity. Longer treatment period.	[167]
BPA, E2, EE2/sediment	Adsorption (aquatic colloids and sediment in a single and binary system).	Equilibrium conc.: 0.40–2.00 mg/L; aquatic colloids: 42.0 mg/L, 103.5 mg/L; initial concentration of EDCs: 0.5–2.5; pH: 8.24–8.37.	NA	Sediments enhance contaminants. sorption process by colloids in a binary system.		[168]
BPA, EE2, CytR, 5-Fu, diazinon, cytrabine, caffeine, phenazone, atrazine, 4-NP/hospital wastewater	Adsorption (PAC)	Dosage: 8, 23, 43 (mg/L); PAC doses: 10, 20, and 40 mg/L; initial conc.: 20, 40, and 80 mg/L. Retention time = 2 days.	The effluent of the PAC reactor was filtered via a flat sheet UF membrane.	Removal efficiencies of diclofenac and carbamazepine and propranolol were 99%, 100%, and ˃94%.	PAC could not remove antibiotic resistance and failed to deactivate pathogens. Energy-intensive.	[25]
Tonalide, BPA, TCS, metolachlor, ketoprofen, and E3/aqueous solutions	Adsorption using PVP-coated magnetite nanoparticles sorbent	pH: 7.5; contact time: 5–40 min; adsorbent dose: 0.75 to 2.5 mg/L; stirring speed: 150 rpm.	NPs were added to the solution followed by sonication. Vials were agitated at 150 rpm. Sample analysis.	The maximum adsorption capacities of BPA and ketoprofen were 90.91 and 83.33 µg/mg, respectively.	NA	[169]
PFOA, PFOS, ACE, DIF, and CHL/eenvironmental water	Batch adsorption (magnetic nanoparticles-attached fluorographene-based sorbent)	Initial conc. of adsorbate: 180 µg/L; adsorbent dose: 400 mg/L; speed: 220 rpm; contact time: 10, 30 min	Solution stirring with developed sorbents and PAC, followed by separation. Measurement of residual EDCs conc.	DIF, ACE, and CHL (97–99%), PFOA removal ranged between 92% and 95%, PFOS (94–97%).	NA	[62]

PFOA, perfluorooctanoic acid; E1, estrone; ACE, acetochlor; E2, 17β-estradiol; PFOS, perfluorooctane sulfonate; EE2, 17α-ethinyl estradiol; CHL, chlorantraniliprole; E3, estriol; DIF, difloxacin hydrochloride; BPA, bisphenol A; OMPs, organic micropollutants; NP, nonylphenol; IBF, ibuprofen; TCS, triclosan; SMZ, sulfamethoxazole; 4 tOP, 4-tert-octyl-phenol; GAC, granular activated carbon; CBZ, carbamazepine; PAC, powdered activated carbon; PVP, polyvinylpyrrolidone; NA, not available.

**Table 7 polymers-13-03229-t007:** Removals of some EDCs using hybrid processes.

Major Contaminants/Sources	Treatment Process	Treatment Factor	Brief Procedure	Major Findings	Limitations	References
E1, BPA, E3, EE2, E2/ secondary effluent	Combined UF and ozonation process	Temperature: 20 ± 2 °C; flux: 28 LMH; O3 dosage: 1.86 mg/L; aeration: 15 min O; EDCs concentration: 50 µg/L; dilution ratios for estrogenicity (EEQC) removal efficiencies: A:100, B:200, C:400.	Adequate mixing of the secondary effluent with EDCs stock solution in the feed tank. Pumping the mixed wastewater into the O3 column. Treatment of the effluent from the UF module and backwashing.	Higher flux (28.9 LMH). Average EDCs removal = 75.2%. Estrogenicity removal = average 84.3%.	Longer operating time (52 days). Fouling. The high cost of ozone production. Generation of solid by-products.	[38,148]
E1, E2, EE2, E3, BPA, 4- NP/influent from WWTFs	Activated sludge, constructed wetland stabilization pond, micro-power biofilm reactor (MP)	Seasonal variation.	Sample collection and analysis. sludge screening. Primary sedimentation. Anoxic aerobic sedimentation. Coarse screening. Stabilization pond.	Higher removal (˃70%) for E1, EE2, NP, and BPA in the activated sludge process in summer.	Stabilization pond was not effective in removing target EDCs such as E1, EE2, and BPA (18–46%) in winter. Poor EDCs removal.	[62]
BPA, NP, E1, E2, EE2, E3, 4NP,4-tert-OP/micro-polluted surface water	Coagulation, ozonation, ceramic membrane UF, and GAC filtration	pH: 6.4–8.5; temp.: 4.1–18.1 °C; conductivity: 239–274 µS/cm.	Pre-oxidation. Coagulation. Sedimentation. Sand filtration. Ozonation. GAC filtration. UF membrane. Disinfection.	Removal efficiencies of target compounds increase from 64% to 100%.	The removal rate of sulfapyridine and BPA were 16% and 7%.	[189]
Acetaminophen, metoprolol, caffeine, antipyrine, SFZ, flumequine, ketorolac, atrazine, isoproturon, 2-hydroxyphenyl, and DFC/wastewater effluent	Adsorption, coagulation, and UF membrane processes	TMP: 4 bar; pH: 8.0; temp: 20 °C; initial conc.: 0–2.0 mg/L; PAC: 10–100 mg/L; adsorption/UF/PAC conc.: 10–600 mg/L; time: 24 h, 120 rpm, and 20 °C. Coagulant dose: 10–20 mg/L. PAC: 0, 10, 20, 50 mg/L.	Soaking new membrane in ultra-pure water for 24 h. Determining the pure water permeate of the membrane by measuring the pure water flux (J_wi_) Filtration of selected contaminants. Analysis of samples.	The combination of PAC adsorption/UF is a promising option.	The amount of PAC (600 mg/L) required for a complete elimination of the selected contaminants is not economical or feasible for the pilot scale.	[190]
EE2/aqueous solutions	AC adsorption and (PAC/UF membrane)	PAC dosage: 0–10 mg/L. Dosage of PAC: 10 mg/L. Filtration rate: 6, 10, 12, 15 L/min.		The removal rate of EE2 ranges from 7.01% to 80.03%.	Separation of PAC from treated effluent remains a technical challenge.	[191]
E1, E2, EE2, E3, and BPA/secondary effluent discharged from WWTP	Ozonation and UF technologies	UVA: 254, 258, 260, 280. Average membrane flux: 23 L (m^2^/h) and 32.3 L (m^2^/h)	Feeding the secondary effluent into an O_3_ reaction tank. Introduction of high purity nitrogen to blow O_3_. Filtration of secondary effluent into the UF.	Ozone (O_3_) & UF recorded almost 100% EDCs removal.	The high cost of ozone production. Production of bromate ions, organic and solid by-products.	[192,193]
BPA, DFC, CBZ, gemfibrozil, naproxen atrazine, TCS/raw water sources	Conventional treatment, ozone/H_2_O_2_ and UV/H_2_O_2_ (both)	Ozone dose: 0.80–4.4 mg/L; ratio of H_2_O_2_/ozone: 0.10 (mass-based); coagulant dose: 10–15 mg/L.		Ozone/H_2_O_2_ and UV/H_2_O_2_ and conventional treatment removed EDCs (97%). Complete removal of TCS, BPA, and DFC. Removals of CBZ, fluoxetine, IBF, naproxen, and atorvastatin were 86–98%.	Conventional treatment poorly removes EDCs. Complex procedure.	[194]
EE2, E2, and E3/biotreated domestic wastewater	AOP and RO	H_2_O_2_ (mg/L) doses: 4, 10, 16. UV dose (kJm^−2^): 24.48–122.4. pH = 4.1–5.9. Turbidity (NTU): 0.54–9.7; TOC (mg/L): 4.1–8.8.	Membrane process. Secondary tank. WWTP. RO process. AOP process.	Higher removal with UV/H_2_O_2_ reaching 91% for E3 and 100% for E2, EE2, respectively.	Advanced oxidative processes are expensive. Potential bromated by-products. Longer filtration time.	[45,195]
EDCs, herbicide, pesticide, DBP volatile/microfiltered effluent	Advanced oxidation UV irradiation and (UV/H_2_O_2_) incorporating MF and RO.	H_2_O_2_ dose (mg/L): 3, 9.5, 16; UV–T (%): 65, 82, 98; UV dose (mJ·cm^−2^): 2, 3, 6, 9, 12, 20; flow rates: 1–3 m^3^h^−1^; retention time: 120–180 s.		AOP achieved significant removal (>90%) for all waters. >99% of estrogenic compounds were removed to 1 ng L^−1^.	The practical limitation exists. The cost penalty is significant. High energy demand.	[196]
EDCs (Food additives, personal care, medicament, industrial additives)/Effluent from WWTP	Biological and continuous mode UF treatment	Ammonium concentration: 30–45 g NH^4+^/m^3^. Pressure: 1.5–2.3 bar. Constant flow: 3.3 m^3^/h. MWCO: 100 kDa. Sampling hours (9:00, 10:00, 11:00, and 12:00 a.m.)	Analyzed samples were collected after initial treatments. Collection of composite grab samples. Freezing of all samples. Samples were collected and filtered through a 1.2 µm glass microfiber filter. Sample analysis.	Compounds showed removal efficiencies >99.5%.	Removal rates of UF treatment were low (<30%) in most cases. Few compounds showed removal rates <20%. Fouling.	[8]

E1, estrone; IBF, ibuprofen; EE2, 17α-ethinyl estradiol; E3, estriol; DFC, diclofenac; BPA, bisphenol A; SMZ, sulfamethaxole; NP, nonylphenol; OP, octyl-phenol; TCS, triclosan; E2, 17β-estradiol; 4 tOP, 4-tert-octyl-phenol; CBZ, carbamazepine; UF, ultrafiltration; NF, nanofiltration; MBR, membrane bioreactor; GAC, granular activated carbon; PAC, activated carbon; AOP, advanced oxidation process; MF, microfiltration; RO, reverse osmosis; CAS, conventional activated sludge.

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
