# Peer review of "Contemporary Techniques for Remediating Endocrine-Disrupting Compounds in Various Water Sources: Advances in Treatment Methods and Their Limitations"

_polymers, 2021, doi:10.3390/polym13193229_

Round 1

Reviewer 1 Report

One of the increasing problems of human health all over the world is fast spread of diabetes and other endocrine disorder related problems. One of the main source of such a problem is water pollution by related contaminations. There are lots of efforts researches do to investigate the issue and find out efficient ways of water purification from endocrine-disrupting compounds. These works encompass a variety of methods (from mechanical to molecular level). However, it seems that there is so far no an analytical work combining these methods in one flask and analyzing them against each other. In relation to this, the present work is quite in need. It critically describes lots of techniques, their applicability, strengths and limitations, the type and quality of information obtained when used separately and together. It would be a useful guide for specialists working in this field.
The strength of the work: The work not only comprehensively describes the past and ongoing situation, but substantiates possible future ways of development of the field under consideration.
The weakness: The work would gain if there were description of a bit more results of representative specific case studies.​
In general, the work is of quite high quality, logically well-arranged and clear, accurately typeset, reference list is comprehensive, up-to-dated and quite complete. In my view, the manuscript is suitable for publication in Polymers in its present form. ​

Author Response

Dear sir,

Kindly find the attachment as per response to the reviewer's comments

Reviewer 2 Report

Manuscript: Contemporary Techniques for Remediating Endocrine Disrupting Compounds in Various Water Sources: Advances in Treatment Methods and their Limitations

The manuscript presents very good work related water purification and going to be interesting for the readers.

Some minor comments are as follows.

  1. Authors need to include some interesting data in the abstract part of the manuscript.
  2. English must be improved.
  3. Novelty of the work be established.
  4. All the important results reported be compared in a tabular form to establish the superiority of the work.
  5. Authors need to add future prospective of presented research in the conclusion part of the manuscript.
  6. Authors must need to incorporate following recent reference related to water purification the manuscript to make it more interesting for the readers.

  • Environmental Science: Water Research & Technology 6 (11), 3080-3090
  • Advanced Sustainable Systems, 1900114
  • ACS omega 4 (26), 22008-22020
  • ACS Sustainable Chemistry & Engineering 7 (6), 6140-6151
  • ACS Sustainable Chemistry & Engineering 6 (3), 3279-3290

Author Response

Dear sir, 

See the attachment for your response. 

All the references recommended has been cited in the manuscrpt.

Thank you.
